# Comparing the evolutionary dynamics of predominant SARS-CoV-2 virus lineages co-circulating in Mexico

**Hugo G Castelán-Sánchez[1,2†], Luis Delaye[1,3†], Rhys PD Inward[4†], Simon Dellicour[5,6†], Bernardo Gutierrez[1,4†], Natalia Martinez de la Vina[4], Celia Boukadida[1,7], Oliver G Pybus[4,8], Guillermo de Anda Jáuregui[1,2,9], Plinio Guzmán[10], Marisol Flores-Garrido[11,12], Óscar Fontanelli[1,3], Maribel Hernández Rosales[1,3], Amilcar Meneses[11,12], Gabriela Olmedo-Alvarez[1,3], Alfredo Heriberto Herrera-Estrella[1,13], Alejandro Sánchez-Flores[1,14], José Esteban Muñoz-Medina[1,15], Andreu Comas-García[1,16], Bruno Gómez-Gil[1,17], Selene Zárate[1,18], Blanca Taboada[1,19], Susana López[1,19], Carlos F Arias[1,19], Moritz UG Kraemer[1,4], Antonio Lazcano[1,20], Marina Escalera Zamudio[1,4]***

[1]Consorcio Mexicano de Vigilancia Genómica (CoViGen-Mex), Mexico City, Mexico; [2]Programa de Investigadoras e Investigadores por México, Consejo Nacional de Ciencia y Tecnología, Mexico City, Mexico; [3]Departamento de Ingeniería Genética, CINVESTAV-Unidad Irapuato, Guanajuato, Mexico; [4]Department of Biology, University of Oxford, Oxford, United Kingdom; [5]Spatial Epidemiology Lab (SpELL), Université Libre de Bruxelles, Bruxelles, Belgium; [6]Department of Microbiology, Immunology and Transplantation, Rega Institute, KU Leuven, Leuven, Belgium; [7]Centro de Investigación en Enfermedades Infecciosas, Instituto Nacional de Enfermedades Respiratorias, Mexico City, Mexico; [8]Department of Pathobiology, Royal Veterinary College, London, United Kingdom; [9]Instituto Nacional de Medicina Genómica, Mexico City, Mexico; [10]Astronomer LTD, Mexico City, Mexico; [11]Escuela Nacional de Estudios Superiores, Universidad Nacional Autónoma de México, Mexico City, Mexico; [12]Departamento de Ciencias de la Computación, CINVESTAV-IPN, Mexico City, Mexico; [13]Laboratorio de expresión génica y desarrollo en hongos, CINVESTAV-Unidad Irapuato, Irapuato, Mexico; [14]Unidad Universitaria de Secuenciación Masiva y Bioinformática, Instituto de Biotecnología, Universidad Nacional Autónoma de México, Chamilpa, Mexico; [15]Coordinación de Calidad de Insumos y Laboratorios Especializados, Instituto Mexicano del Seguro Social, Mexico City, Mexico; [16]Facultad de Medicina y Centro de Investigación en Ciencias de la Salud y Biomedicina, Universidad Autónoma de San Luis Potosí, San Luis Potosí, Mexico; [17]Centro de Investigación en Alimentación y Desarrollo-CIAD, Unidad Regional Mazatlán en Acuicultura y Manejo Ambiental, Sinaloa, Mexico; [18]Posgrado en Ciencias Genómicas, Universidad Autónoma de la Ciudad de México, Mexico City, Mexico; [19]Departamento de Genética del Desarrollo y Fisiología Molecular, Universidad Nacional Autónoma de México, Cuernavaca, Mexico; [20]Facultad de Ciencias, Universidad Nacional Autónoma de Méxic, Mexico City, Mexico

*For correspondence:
marina.escalerazamudio@zoo.ox.ac.uk

†These authors contributed equally to this work

**Competing interest:** The authors declare that no competing interests exist.

**Abstract** Over 200 different SARS-CoV-2 lineages have been observed in Mexico by November 2021. To investigate lineage replacement dynamics, we applied a phylodynamic approach and explored the evolutionary trajectories of five dominant lineages that circulated during the first year

of local transmission. For most lineages, peaks in sampling frequencies coincided with different epidemiological waves of infection in Mexico. Lineages B.1.1.222 and B.1.1.519 exhibited similar dynamics, constituting clades that likely originated in Mexico and persisted for >12 months. Lineages B.1.1.7, P.1 and B.1.617.2 also displayed similar dynamics, characterized by multiple introduction events leading to a few successful extended local transmission chains that persisted for several months. For the largest B.1.617.2 clades, we further explored viral lineage movements across Mexico. Many clades were located within the south region of the country, suggesting that this area played a key role in the spread of SARS-CoV-2 in Mexico.

## Editor's evaluation

Castelán-Sánchez et al. analyzed, in an important way, the SARS-CoV-2 genomes from Mexico collected between February 2020 and November 2021. This period spans three major spikes in daily COVID-19 cases in Mexico and the rise of three distinct variants of concern (VOCs; B.1.1.7, P.1., and B.1.617.2). The authors perform convincing appropriate methodology in line with the current state of the art by careful phylogenetic analyses of these three VOCs, as well as two other lineages that rose to substantial frequency in Mexico, focusing on identifying periods of cryptic transmission (before the lineage was first detected) and introductions to and from the neighboring United States.

## Introduction

Genome sequencing efforts for the surveillance of the Severe Acute Respiratory Syndrome Coronavirus-2 (SARS-CoV-2) has granted public access to a massive number of virus genomes generated worldwide (https://www.gisaid.org/). Exploring SARS-CoV-2 genome data using genomic epidemiology has allowed researchers to characterize increasing virus diversity (*Hill et al., 2021*), track emerging viral subpopulations, and explore virus evolution in real-time, both at local and global scales (for examples see *Kraemer et al., 2021*; *du Plessis et al., 2021*; *Worobey et al., 2020*; *Candido et al., 2020*; *COVID-19 Genomics UK (COG-UK) consortiumcontact@cogconsortium.uk, 2020*). Throughout the development of the COVID-19 pandemic, viral variants have emerged and circulated across different regions of the world (*Kraemer et al., 2021*; *Chen et al., 2021*), displaying specific mutations that define their phylogenetic patterns (*Li et al., 2021*; *Tao et al., 2021*). The emergence and spread of SARS-CoV-2 lineages has been routinely monitored since early 2021, informing public health authorities on their responses to the ongoing pandemic (*Oude Munnink and Koopmans, 2023*).

Emerging virus lineages are classified using a dynamic nomenclature system ('Pango system', Phylogenetic Assignment of Named Global Outbreak Lineages), developed to consistently assign newly generated genomes to existing lineages, and to designate novel virus lineages according to their phylogenetic identity and epidemiological relevance (*Rambaut et al., 2020*; *Ruis and Colquhoun, 2021*). Virus lineages that may pose an increased risk to global health have been classified as Variants of Interest (VOI), Variants under Monitoring (VUM), and Variants of Concern (VOC), potentially displaying one or more of the following biological properties (*Tao et al., 2021*; *Centers for Disease Control and Prevention, 2022*): increased transmissibility (*Horby et al., 2021*), decreasing the effectiveness of available diagnostics or therapeutic agents (such as monoclonal antibodies) (*Weisblum et al., 2020*), and evasion of immune responses (including vaccine-derived immunity) (*Oude Munnink and Koopmans, 2023*; *Greaney et al., 2021a*; *Greaney et al., 2021b*). Up to date, five SARS-CoV-2 lineages (including all descending sub-lineages) have been designated as VOC: B.1.1.7 (Alpha), B.1.351 (Beta), P.1 (Gamma), B.1.617.2 (Delta), and B.1.1.529 (Omicron) (*Oude Munnink and Koopmans, 2023*; *World Health Organization, 2021*; *O'Toole et al., 2021a*).

Virus lineages that dominate across various geographic regions are likely to have an evolutionary advantage, driven in part by a genetic increase in virus fitness (i.e. mutations enhancing transmissibility and/or immune escape; *Martin et al., 2021*; *Escalera-Zamudio et al., 2023*; *Kumar et al., 2021*; *Vöhringer et al., 2021*; *Caniels et al., 2021*). Moreover, the spread of different VOC across the world has been linked to human movement, often resulting in the replacement of previously dominating virus lineages (*Oude Munnink and Koopmans, 2023*). However, exploring lineage replacement/fitness dynamics remains a challenge, as these are impacted by numerous factors, including differential and stochastic growth rates that vary across geographic regions, a shifting immune structure of

the host population (linked to viral pre-exposure levels and vaccination rates), (*Vöhringer et al., 2021*; *Caniels et al., 2021*) and changing social behaviours (such as fluctuating human mobility patterns and the implementation of local non-pharmaceutical interventions across time) (*Vöhringer et al., 2021*; *Zhang et al., 2020*; *Boutin et al., 2021*). Thus, the epidemiological and evolutionary processes enabling some lineages to spread and become dominant across distinct geographic regions, whilst others fail to do so, remain largely understudied.

Mexico has been severely impacted by the COVID-19 pandemic, evidenced by a high number of cumulative deaths relative to other countries in Latin America (*Sánchez-Talanquer et al., 2021*). Since the first introductions of the virus in early 2020 and up to November 2021 (*Taboada et al., 2020*), the local epidemiological curve fluctuated between three waves of infection (observed in July 2020, January 2021, and August 2021; *Sánchez-Talanquer et al., 2021*; *Taboada et al., 2020*; *Ritchie, 2020*). Prior to the first peak of infection, non-pharmaceutical interventions (including social distancing and suspension of non-essential activities) were implemented at a national scale from March 23, 2020 to May 30, 2020. Nonetheless, a reopening plan for the country was already announced in May 13, 2020, whilst the national vaccination campaign did not begin before December 2020 (*Sánchez-Talanquer et al., 2021*). The 'Mexican Consortium for Genomic Surveillance' (abbreviated CoViGen-Mex) *Consorcio de Vigilancia Genómica MexCoV2, 2021a* was launched in February 2021, establishing systematic sequencing effort for a genomic epidemiology-based surveillance of SARS-CoV-2 in Mexico. In close collaboration with the national ministry of health, and driven by the sequencing capacity in the country, the program aimed to sequence per month approximately 1200 representative samples derived from positive cases recorded throughout national territory, based on the proportion of cases reported across states. In May 2021, the sequencing scheme was upgraded to follow the official case report line, in order to better coordinate case reporting and genome sampling across the country.

Derived from publicly accessible genome data from Mexico deposited in GISAID (https://www.gisaid.org/) from 2020 to 2021 (corresponding to the first year of the epidemic), over 48,000 viral genomes were available, resulting in an approximate average of 2000 viral genomes sequenced per month. However, starting February 2021, the CoViGen-Mex sampling scheme gradually increased its sequencing output from a mean of less than 500 genomes per month to over 1000 genomes from May 2021 onwards. As of the time of writing this manuscript, around 80,000 SARS-CoV-2 genomes from Mexico were available in GISAID, with one-third of these generated by CoViGen-Mex *Consorcio de Vigilancia Genómica MexCoV2, 2021a* (and other national institutions sequencing the rest). From these, approximately 95% of all SARS-CoV-2 genomes from Mexico were generated in the country and 5% in the USA.

Investigating the dominance and replacement patterns of SARS-CoV-2 can provide valuable information for understanding viral spread and shed light on virus evolution and adaptation processes. During the first year of the epidemic in Mexico, over 200 different virus lineages were detected, including all VOCs *O'Toole et al., 2021a*; *Consorcio de Vigilancia Genómica MexCoV2, 2021b*. Various virus lineages co-circulated across the national territory, a noteworthy observation in light of recombinant SARS-CoV-2 lineages that emerged in North America during 2021 (*Gutierrez et al., 2022*). Additionally, some virus lineages displayed specific dominance and replacement patterns that differed from those observed in neighboring countries, specifically the USA (*Consorcio de Vigilancia Genómica MexCoV2, 2021a*; *Consejo Nacional de Humanidades, Ciencias y Tecnologías, 2021*; *Taboada et al., 2021*). With this in mind, our study aimed to examine the dominance and replacement patterns of SARS-CoV-2 in Mexico from 2020 to 2021. We explored whether the spread of dominant lineages was driven by specific mutations that impacted local growth rates (further shaped the immune landscape of the local host population, depending mostly on virus pre-exposure levels at this time). We also investigated whether viral diffusion processes within the country were associated with local human mobility patterns, anticipating that the SARS-CoV-2 epidemic in Mexico may have been impacted by the epidemiological behavior of the virus in neighboring countries.

To achieve this, we investigated the introduction, spread, and replacement dynamics of five virus lineages that dominated during the first year of the epidemic: B.1.1.222, B.1.1.519, B.1.1.7 (VOC Alpha), P.1 (VOC Gamma), and B.1.617.2 (VOC Delta) (*Consorcio de Vigilancia Genómica MexCoV2, 2021a*; *Consejo Nacional de Humanidades, Ciencias y Tecnologías, 2021*; *Taboada et al., 2021*). We used a phylodynamic approach to analyze publicly available cumulative SARS-CoV-2 genome data

from the country in the context of virus genome data collected worldwide. We also devised a human migration and phylogenetic-informed subsampling approach to increase the robustness of our tailored phylogeographic analyses. To investigate lineage-specific spatial epidemiology, we contrasted our phylodynamic results with epidemiological and human mobility data, focusing on quantifying lineage importations into Mexico and characterizing local extended transmission chains across geographic regions (i.e. states). Our analysis revealed similar dynamics for the B.1.1.222 and B.1.1.519 lineages, with both likely originating in Mexico and denoting single extended transmission chains sustained for over a year. For P.1, B.1.1.7, and B.1.617.2 lineages, multiple introduction events were identified, with the detection of a few large transmission chains across the country. For B.1.617.2, which represented the largest and most genetically diverse clades identified, we observed a within-the-country virus diffusion pattern seeding from the south with subsequent movement into the central and north. We further identify Mexico's southern border may have played an important role in the introduction and spread of SARS-CoV-2 (and other epidemics) across the country.

## Results

The sampling date of this study comprises January 2020 to November 2021, corresponding to the first year of the epidemic in Mexico, just before the introduction of 'Omicron' (B.1.1.529) into the country (*Consorcio de Vigilancia Genómica MexCoV2, 2021a*; *Consejo Nacional de Humanidades, Ciencias y Tecnologías, 2021*; *Taboada et al., 2021*). During this time, Mexico reported a daily mean test rate ranging between 0.13–0.18 test per 1,000 inhabitants (*Ritchie, 2020*). Despite a lower testing rate compared to other countries, the cumulative number of viral genomes generated throughout 2020 and 2021 (both by CoViGen-Mex and other national institutions) correlates with the number of cases recorded at a national scale, corresponding to approximately 100 viral genomes per 10,000 cases, or ~1% of the official COVID-19 cases (*Figure 1—figure supplement 1*). Although SARS-CoV-2 sequencing remained centralized to Mexico City, the proportion of viral genomes per state roughly coincided with the spatial distribution of confirmed cases (with Mexico City reporting most cases), as stated officially *Informe Técnico Diario COVID-19 MÉXICO, 2021* (*Figure 1—figure supplement 1*). Therefore, SARS-CoV-2 sequencing in Mexico has been sufficient to explore the spatial and temporal frequency of viral lineages across national territory (*Consorcio de Vigilancia Genómica MexCoV2, 2021a*; *Consejo Nacional de Humanidades, Ciencias y Tecnologías, 2021*; *Taboada et al., 2021*), now to further investigate the number of lineage-specific introduction events, and to characterize the extension and geographic distribution of associated transmission chains under a genomic epidemiology approach, as presented in this study. Our comparative analysis on the temporal distribution of virus lineages in Mexico confirmed previous published observations (*O'Toole et al., 2021a*; *Consejo Nacional de Humanidades, Ciencias y Tecnologías, 2021*; *Taboada et al., 2021*) showing that relative to other virus lineages circulating at the time, only the B.1.1.222, B.1.1.519, B.1.1.7 (Alpha), P.1 (Gamma), and B.1.617.2 (Delta) lineages displayed a dominant prevalence pattern within the country. Moreover, for most of these dominant lineages, peaks in genome sampling frequency (defined here as the proportion of viral genomes assigned to a specific lineage, relative to the proportion of viral genomes assigned to any other virus lineage in a given time point) often coincided with the epidemiological waves of infection recorded (except for B.1.1.7 and P.1) (*Figure 1a and b*).

### B.1.1.222

The B.1.1.222 lineage circulated in North America between April 2020 and September 2021, mostly within the USA (~80% of all B.1.1.222-assigned genomes) and Mexico (~20% of all B.1.1.222-assigned genomes). With limited reports from other regions of the world, B.1.1.222 was thus considered as endemic to the region (https://cov-lineages.org/) *O'Toole et al., 2021b*. The first B.1.1.222-assigned genome was sampled from Mexico on April 2020 (Mexico/CMX-INER-0026/2020-04-04) *O'Toole et al., 2021b*, whilst the last B.1.1.222-assigned genome was sampled from the USA on September 2021 (USA/CA-CDPH-1002006730/2021-09-14). The latest sampling date for B.1.1.222 in Mexico corresponds to July 2021 (Mexico/CHH_INER_IMSS_1674/2021-07-26), two months before the latest sampling date of the lineage at an international scale. We observe that in Mexico, the B.1.1.222 lineage was continuously detected between April 2020 and May 2021, followed by a steady decline after July 2021 (*Figure 1b*). During its circulation period, most B.1.1.222 genomes were collected

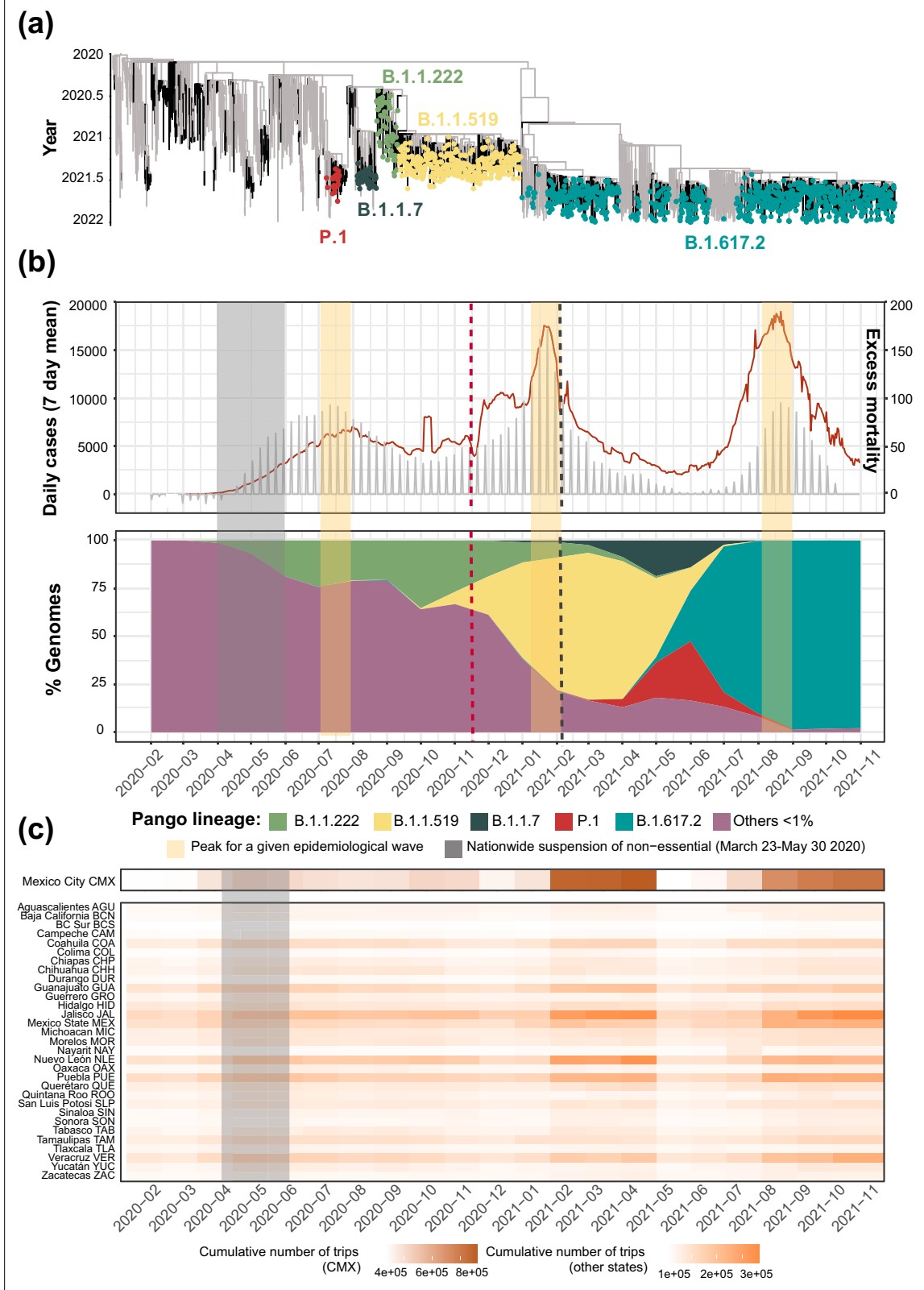

**Figure 1.** Overview of the SARS-CoV-2 epidemic in Mexico. (**a**) Time-scaled phylogeny of representative SARS-CoV-2 genomes from Mexico within a global context, highlighting the phylogenetic positioning of B.1.1.222, B.1.1.519, B.1.1.7, P.1, and B.1.617.2 sequences. Lineage B.1.1.222 is shown in light green, B.1.1.519 in yellow, P.1 in red (Gamma), B.1.1.7 (Alpha) in dark green, and B.1.617.2 (Delta) in teal (**b**) The epidemic curve for COVID-19 in Mexico from January 2020 up to November 2021, showing the average number of daily cases (red line) and associated excess mortality (represented

*Figure 1 continued on next page*

Figure 1 continued

by a punctuated grey curve, denoting weekly average values). The peak of the first (July 2020), the second (January 2021), and the third wave (August 2021) of infection are highlighted in yellow shadowing. The dashed red line corresponds to the start date national vaccination campaign (December 2020), whilst the dashed black line represents the implementation date of a systematic genome sampling and sequencing scheme for the surveillance of SARS-CoV-2 in Mexico (February 2021). The period for the implementation of non-pharmaceutical interventions at national scale is highlighted in grey shadowing. The lower panel represents the genome sampling frequency (defined here as the proportion of viral genomes assigned to a specific lineage, relative to the proportion of viral genomes assigned to any other virus lineage in a given time point) of dominant virus lineages detected in the country during the first year of the epidemic. Lineages displaying a lower sampling frequency are jointly shown in purple. (**c**) Heatmap displaying the volume of trips into a given state from any other state recorded from January 2020 up to November 2021 derived from anonymized mobile device geolocated and time-stamped data.

The online version of this article includes the following figure supplement(s) for figure 1:

**Figure supplement 1.** Cumulative number of genome sequences generated per state (data available up to March 2022).

**Figure supplement 2.** Mean interstate connectivity recorded between 2021 and 2022.

from the central region of the country, represented by Mexico City (CMX; *Figure 2a*). For B.1.1.222, a rising genome sampling frequency was observed from May 2020 onwards, coinciding with the first epidemiological wave recorded during July 2020. Subsequently, genome sampling frequency progressively increased to reach a highest of 35% recorded in October 2020, denoting established dominance before the emergence and spread of B.1.1.519 (*Figure 1b*).

Data from the first year of the epidemic (available until February 2021, as analyzed by *Taboada et al., 2021*; *O'Toole et al., 2021a*; *Consejo Nacional de Humanidades, Ciencias y Tecnologías, 2021*) initially revealed that the B.1.1.222 lineage had reached a maximum genome sampling frequency of approximately 10%. However, our results show an important frequency underestimation (10% vs 35%), since the vast majority of B.1.1.222-assigned genomes from Mexico (>80%) were generated, assigned, and submitted to GISAID after February 2021. These observations are based on publicly available genome data, and both these values may actually underestimate lineage prevalence. However, calculating 'real' frequency values goes beyond the scope of this study. Notably, in the USA, the B.1.1.222 lineage reached a maximum genome sampling frequency of 3.5%, compared to a 35% observed in Mexico. This four-fold difference in the number of B.1.1.222-assigned genomes between the USA and Mexico reflects a significant disparity in sequencing efforts between the two countries, and contrasts with region-specific epidemiological scenarios *Hill et al., 2021*. This is of great importance since sequencing disparities and sampling gaps between countries can hinder the development of global outbreak control strategies and exacerbate existing health inequalities.

Phylodynamic analysis for the B.1.1.222 lineage revealed one main clade deriving from a single earliest MRCA (most recent common ancestor) with a 'most likely' location (supported by a relative Posterior Probability [PP] of 0.99) inferred to be 'Mexico', denoting lineage emergence in the country (*Figure 2a*). The inferred date for this MRCA corresponds to March 2020, further denoting a cryptic circulation period of a month (before the earliest sampling date for the lineage within the country, see Methods section "Time-scaled analysis"). Subsequent 'introductions' should be interpreted as 're-introduction' events into the country (with dates ranging from October 2020 to July 2021). After emergence, B.1.1.222 was seeded into the USA from Mexico multiple times. In this context, we estimate a minimum of 237 introduction events from Mexico into the USA (95% HPD interval = [225-250]), and a minimum of 106 introduction events from the USA into Mexico (95% HPD interval = [93-120]; *Figure 2a*). Based on inferred node dates (for MRCAs) in the MCC tree, the B.1.1.222 lineage displayed a total persistence of up to 16 months.

## B.1.1.519

Directly descending from B.1.1.222 (*Figure 1a*), the B.1.1.519 lineage circulated in North America between August 2020 and November 2021, mostly within the USA (~60% of all B.1.1.519-assigned genomes) and Mexico (~30% of all B.1.1.519-assigned genomes). As for B.1.1.222, B.1.1.519 genome reporting from other countries was limited, and the B.1.1.519 lineage was also considered as endemic to the region (https://cov-lineages.org/) (*Taboada et al., 2021*; *O'Toole et al., 2021l*; *Cedro-Tanda et al., 2021*; *Rodríguez-Maldonado et al., 2021*). At an international scale, the earliest B.1.1.519-assigned genome was sampled from the USA on July 2020 (USA/TX-HMH-MCoV-45579/2020-07-31) *O'Toole et al., 2021c*, whilst the latest B.1.1.519-assigned genome was sampled from Mexico on

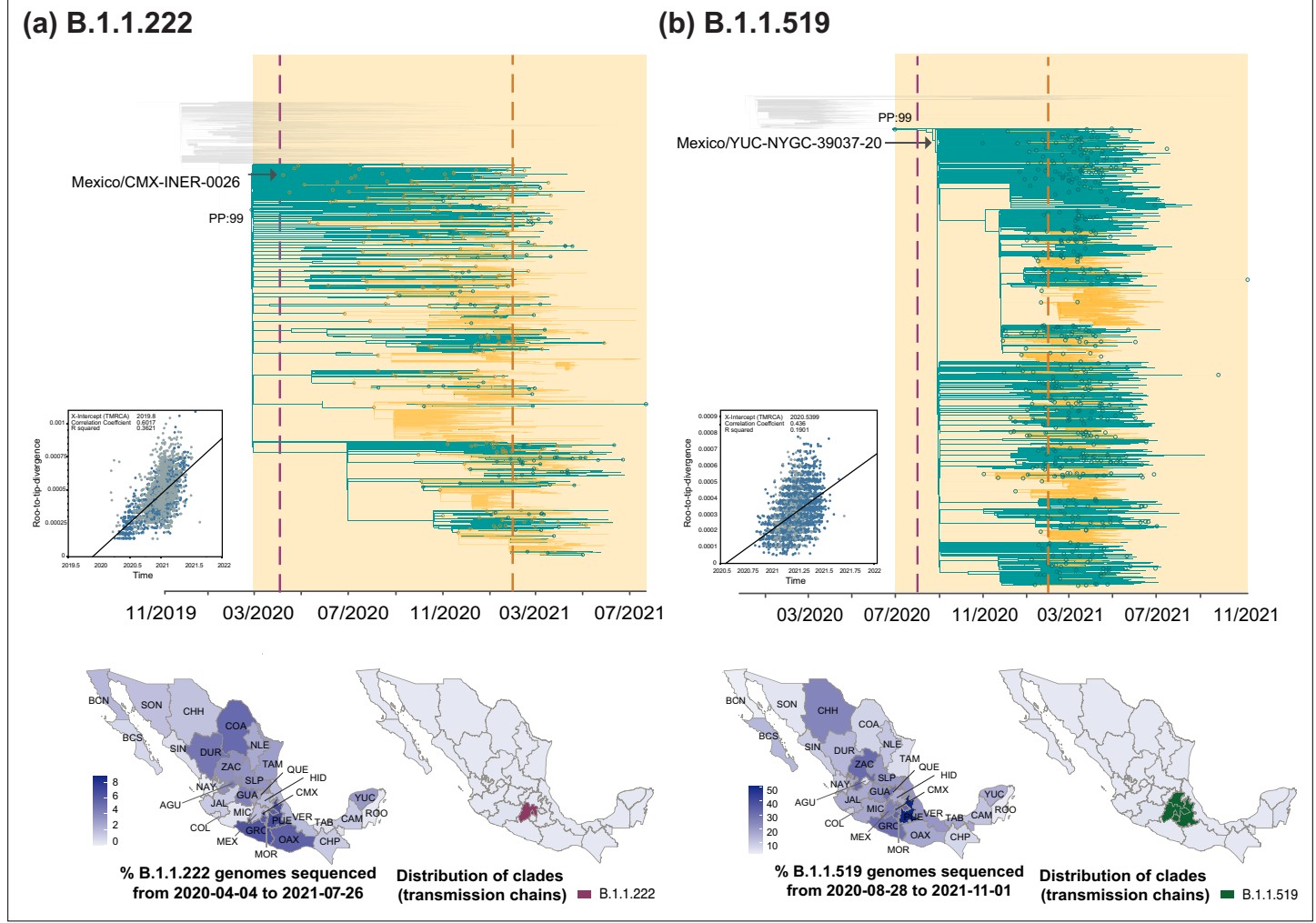

**Figure 2.** Time-scaled phylogenetic analyses for the B.1.1.222 and B.1.1.519 lineage. Maximum clade credibility (MCC) trees for the (**a**) B.1.1.222 and (**b**) B.1.1.519 lineages, in which clades corresponding to distinct introduction events into Mexico are highlighted. Nodes shown as outline circles correspond to the most recent common ancestor (MRCA) for clades representing independent re-introduction events into Mexico (in teal) or from the USA (in ochre). Based on the earliest and latest MRCAs, the estimated circulation period for each lineage is highlighted in yellow shadowing. The dashed purple line represents the date of the earliest viral genome sampled from Mexico, while its position in the tree indicated. The dashed yellow line represents the implementation date of a systematic virus genome sampling and sequencing scheme for the surveillance of SARS-CoV-2 in Mexico. The corresponding root-to-tip regression plots for each tree are shown, in which genomes sampled from Mexico are shown in blue, whilst genomes sampled elsewhere are shown in grey. Map graphs on the left show the cumulative proportion of genomes sampled across states per lineage of interest, corresponding to the period of circulation of the given lineage (relative to the total number of genomes taken from GISAID, corresponding to raw data before subsampling). Maps on the right represent the geographic distribution of the clades identified.

December 2021 (Mexico/CHP_IBT_IMSS_5310/2021-12-27) *Aceves et al., 2021*. During initial phylogenetic assessment, we noted that most of B.1.1.519-assigned genomes collected after November 2021 came from outside North America (namely, from Turkey and Africa). These were further identified as outliers within the tree, likely to be sequencing errors resulting, and thus were excluded from further analyses. In Mexico, the B.1.1.519 lineage was first detected on August 2020 (Mexico/YUC-NYGC-39037-20/2020-08-28) (*Taboada et al., 2021*).

Our analysis derived from cumulative genome data from Mexico shows that B.1.1.519 displayed an increasing genome sampling frequency observed between September 2020 and July 2021 (*Figure 1b*). During these months, the spread of B.1.1.519 raised awareness in public health authorities, leading to its designation as a VUM in June 2021 (*Oude Munnink and Koopmans, 2023*; *Taboada et al., 2021*; *Cedro-Tanda et al., 2021*; *Rodríguez-Maldonado et al., 2021*). During its circulation period, most B.1.1.519 genomes were sampled from the central region of the country, represented by the

state of Puebla (PUE; *Figure 2b*). We further observed that by late January 2021, up to 75% of the virus genomes sequenced in Mexico were assigned as B.1.1.519, with the lineage dominating over the second wave of infection recorded (*Figure 1b*). Similar to B.1.1.222, in the USA, B.1.1.519 only reached a maximum genome sampling frequency of 5% (up to April 2021). Compared to the 75% observed in Mexico, this once again contrast to the epidemiological scenario observed in each country, further exposing sequencing disparities (*O'Toole et al., 2021b*; *O'Toole et al., 2021c*).

Phylodynamic analysis for the B.1.1.519 lineage revealed a similar pattern to the one observed for B.1.1.222, with one main clade deriving from a single MRCA (*Figure 2b*). The inferred date for this MRCA corresponds to July 2020, again with a 'most likely' source location inferred to be 'Mexico' (PP: 0.99). Thus, our results suggest that B.1.1.519 circulated cryptically in Mexico for one month prior to its initial detection (*Figure 2b*). After its emergence, the B.1.1.519 lineage was seeded back and forth between the USA and Mexico, with subsequent 're-introduction events' into the country occurring between July 2020 and November 2021. In this light, we estimate a minimum number of 121 introduction events from the USA into Mexico (95% HPD interval = [108-131]), compared to a minimum number of 391 introduction events from Mexico into the USA (95% HPD interval = [380-402]) (*Figure 2b*). Based on inferred node dates in the MCC tree, the B.1.1.519 lineage displayed a total persistence of over 16 months.

## B.1.1.7

The B.1.1.7 lineage was first detected in the UK in September 2020, spreading to more than 175 countries in over a year *O'Toole et al., 2021d*. The earliest B.1.1.7-assigned genome from Mexico was sampled on late December 2020 (Mexico/TAM-InDRE-94/2020-12-31), while the latest B.1.1.7-assigned genome was sampled on October 2021 (Mexico/QUE_InDRE_FB47996_S8900/2021-10-13). Our analysis derived from cumulative genome data from the country revealed a continuous detection between February and September 2021. A peak in genome sampling frequency was observed around May 2021, coinciding with a lower number of cases recorded at the time (*Figure 1b*). Our results further confirm that the B.1.1.7 lineage reached an overall lower sampling frequency of up to 25% (relative to other virus lineages circulating in the country), as noted prior to this study (e.g. see *Zárate et al., 2022*; *Sánchez-Talanquer et al., 2021*; *Ritchie, 2020*; *Subsecretaria de Prevención y Promoción de la Salud, 2021b*). Of interest, similar observations were independently made for other Latin American countries, such as Brazil, Chile, and Peru (https://www.gisaid.org/), likely denoting region-specific dynamics for this lineage.

Phylodynamic analysis for B.1.1.7 revealed an earliest MRCA dating to late October 2020, denoting a cryptic circulation period of approximately two months prior to detection in the country. The earliest genome sampling date also coincides with at least four independent and synchronous introduction events that date back to December 2020 (*Figure 3a*). In total, we estimated a minimum of 224 introduction events into Mexico (95% HPD interval = [219-231]). Potentially linked to the establishment of a systematic genome sequencing in Mexico, most of these were identified after February 2021. Within the MCC, we further identified seven clades (C1a to C7a) representing extended local transmission chains, with C3 and C7 being the largest (*Figure 3a*, *Supplementary file 2*).

During its circulation period, most B.1.1.7 genomes from Mexico were generated from the state of Chihuahua, with these also representing the earliest B.1.1.7-assigned genomes from the country (*Taboada et al., 2021*; *Zárate et al., 2022*). However, only a small proportion of these genomes grouped within a larger clade denoting an extended transmission chain (C2a), with the rest falling within minor clusters, or representing singleton events (*Figure 3a*). Relative to other states, Chihuahua generated an overall lower proportion of viral genomes throughout 2020–2021 (*Figure 1—figure supplement 1*). Between February 2021 and September 2021 (corresponding to the circulation period of the B.1.1.7 lineage/Alpha VOC in Mexico), Mexico City reported the highest number of COVID-19 cases (https://coronavirus.gob.mx/datos/#DOView). During this time, Mexico City also reported the highest number of cases related to the B.1.1.7 lineage/Alpha VOC, with 2452 confirmed cases, followed by the states of Mexico, Jalisco, and Nuevo Leon (https://coronavirus.gob.mx/variantes/). Therefore, neither phylogenetic nor epidemiological data from the country support the notion that Chihuahua may have been an initial sink-source for the B.1.1.7 lineage/Alpha VOC (or for any other virus lineage, when comparing DTA results). Various factors can impact virus lineage distribution in a given region at a specific time point, including stochastic population processes, and the role

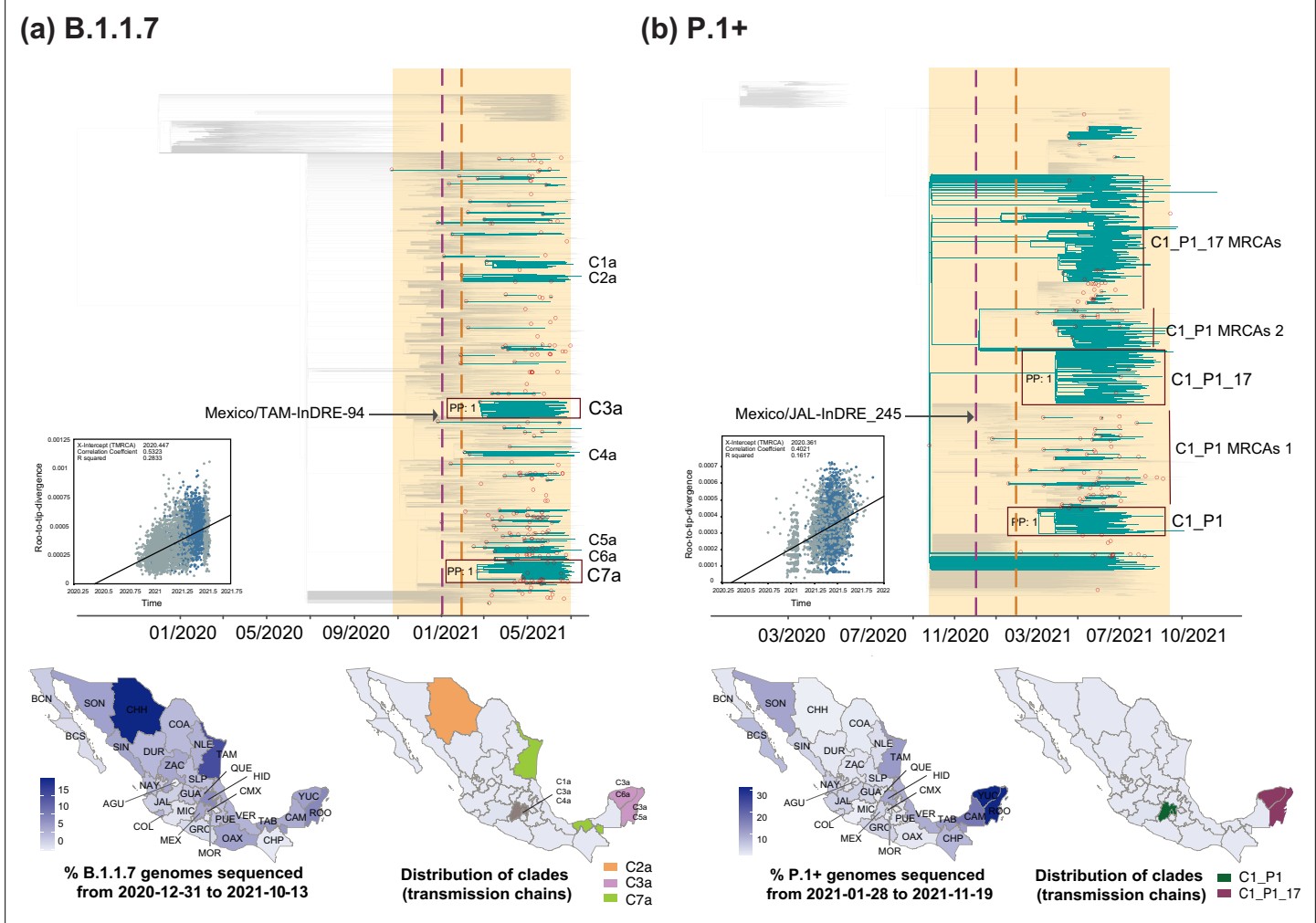

**Figure 3.** Time-scaled phylogenetic analyses for the B.1.1.7 and P.1 lineages. Maximum clade credibility (MCC) trees for the (**a**) B.1.1.7 and the (**b**) P.1 lineages, in which major clades identified as distinct introduction events into Mexico are highlighted. Nodes shown as red outline circles correspond to the most recent common ancestor (MRCA) for clades representing independent introduction events into Mexico. Based on the earliest and latest MRCAs, the estimated circulation period for each lineage is highlighted in yellow shadowing. The dashed purple line represents the date of the earliest viral genome sampled from Mexico, while its position in the tree indicated. The dashed yellow line represents the implementation date of a systematic virus genome sampling and sequencing scheme for the surveillance of SARS-CoV-2 in Mexico. The corresponding root-to-tip regression plots for each tree are shown, in which genomes sampled from Mexico are shown in blue, whilst genomes sampled elsewhere are shown in grey. Map graphs on the left show the cumulative proportion of genomes sampled across states per lineage of interest, corresponding to the period of circulation of the given lineage (relative to the total number of genomes taken from GISAID, corresponding to raw data before subsampling). Maps on the right represent the geographic distribution of the clades identified.

The online version of this article includes the following figure supplement(s) for figure 3:

**Figure supplement 1.** Largest 'Mexico' clades within the B.1.1.7 MCC tree.

**Figure supplement 2.** Largest 'Mexico' clades within the P.1+MCC tree.

of asymptomatic carriers, which can contribute to the difficulty in identifying extended transmission chains and their geographic distribution. Consequently, we can only speculate that given its proximity to the US border, Chihuahua may have been an early introduction point of the lineage from the US. However, this observation is not supported by our phylogeographic analyses, given the restrictions on determining source locations for virus introduction events into the country related to sampling limitations.

For the larger C3a and C7a clades, both MRCA's date to February 2021, denoting independent and synchronous introduction events (*Figure 3a*). The C3a comprises genomes collected from 22/32 states in the country, predominantly from Mexico City (CMX), followed by southern states of Yucatán

(YUC) and Quintana Roo (ROO) (*Figure 3—figure supplement 1*). The C3a displayed a persistence of three months: from March to June 2021. For the C7a, viral genomes were sampled from 20/32 states of the country, with >70% of these coming from the southern state of Tabasco (TAB) and north-eastern state of Tamaulipas (TAM) (*Figure 3—figure supplement 1*). The C7a displayed a persistence of four months: from March to July 2021. Based on inferred node dates within the MCC tree, the B.1.1.7 lineage displayed a total persistence of approximately 10 months.

## P.1

The P.1 lineage was first detected in Brazil during October 2020 (*Faria et al., 2021*), after which it diverged into >20 sub-lineages that spread to different parts of the world *O'Toole et al., 2021a*. Relevant to North America, P.1.17 was the most prevalent sub-lineage detected within the region, again sampled mostly from the USA (~60% of all sequences) and from Mexico (~30% of all sequences, https://cov-lineages.org/) *O'Toole et al., 2021a*. In Mexico, we detected at least 13 P.1 sub-lineages, with the majority of assigned viral genomes belonging to the P.1.17 (66%), and to a lesser extent to the parental P.1 lineage (25%), as was noted prior to this study *Consorcio de Vigilancia Genómica MexCoV2, 2021b*. As our dataset comprises viral genomes assigned to the P.1 and descending sub-lineages, it is henceforth referred here as a P.1+.

The earliest P.1+genome from Mexico was sampled on late January 2021 (Mexico/JAL-InDRE_245/2021-01-28) and the latest on November 2021 (Mexico/ROO_IBT_IMSS_4502/2021-11-19). Cumulative genome data analysis from the country revealed a similar pattern to that observed for B.1.1.7, in which P.1+genome sampling frequency peaked around April-May 2021, with almost no detection after September 2021. As for B.1.1.7, P.1+showed an overall lower genome sampling frequency reaching a highest of 25%, again coinciding with a decrease in the number of cases following the second wave of infection recorded (*Figure 1b*; *Consorcio de Vigilancia Genómica MexCoV2, 2021b*; *Consejo Nacional de Humanidades, Ciencias y Tecnologías, 2021*; *Taboada et al., 2021*). During its circulation period in the country, the majority of P.1+-assigned genomes were sampled from the states Yucatan and Quintana Roo (YUC and ROO; *Figure 3b*).

Our phylodynamic analysis for P.1+revealed a minimum number of 130 introduction events into Mexico (95% HPD interval = [116-140]). Within the MCC tree, we identified two well-supported clades denoting extended local transmission chains: C1_P1 (corresponding to P.1) and C1_P1_17 (corresponding to P.1.17; *Figure 3b*, *Supplementary file 2*). The MRCA of the C1_P1 clade dates to March 2021, showing a persistence of seven months: from March to October 2021. The MRCA of C1_P1_17 dates to October 2020, corresponding to the TMRCA of the global P.1+clade in the MCC tree. The long branch separating this earliest MRCA and the earliest sampled sequence reveals a considerable lag between lineage emergence and first detection, likely resulting from sub-lineage under-sampling (*Figure 3b*). Therefore, it is not possible to estimate a total lineage persistence based on inferred node dates. Thus, considering tip dates only, the C1_P1_17 clade showed a persistence of five months (earliest collection date: 01/04/2021, latest collection date: 17/09/2021). For the P.1 parental lineage, two clusters of MRCAs representing subsequent introduction events with no evidence of extended transmission were identified (referred here as clade C1_P1 MRCAs 1 and 2). Similarly, for the P.1.17, another cluster of MRCAs representing subsequent introduction events with no evidence of extended transmission was also identified (referred here as C1_P1_17 MRCAs; *Figure 3b*).

The C1_P1 clade directly descends from viral genomes sampled from South America, and is mostly represented by viral genomes collected from the central region of the country (>40% of these coming from Mexico City and the State of Mexico; CMX and MEX; *Figure 3—figure supplement 2*). The C1_P1_17 clade is mostly represented by viral genomes from Mexico (75%), and to a lesser extent by genomes from the USA (20%). 'Mexico' genomes are positions basally to the C1_P1_17 clade, collected predominantly from the southern region of the country (>90%, represented by the states of Quintana Roo and Yucatán, ROO and YUC; *Figure 3—figure supplement 2*). Overall, our results indicate that in Mexico, the P.1 parental lineage was introduced independently and later than P.1.17, likely from distinct geographic locations. Contrasting to P.1, the P.1.17 lineage displayed a more successful spread, denoted by a sustained transmission chain located to the southern region of the country.

## B.1.617.2

Initially detected in India during October 2020, the B.1.617.2 lineage spread globally to become dominant, and was later associated with an increase in COVID cases recorded globally following March 2021 (*O'Toole et al., 2021e*; *Tian et al., 2021*). The parental B.1.617.2 lineage further diverged into >230 descending sub-lineages (designated as the AY.X) that spread to different regions of the world (*O'Toole et al., 2021a*; *O'Toole et al., 2021e*; *O'Toole et al., 2021f*). Again, as our dataset comprises both B.1.617.2 and AY. X-assigned genomes, it is henceforth referred here as a B.1.617.2+.

The first 'B.1.617.2-like' genome from Mexico was sampled on September 2020 (Mexico/AGU-InDRE_FB18599_S4467/2020-09-22), followed by a sporadic genome detection throughout January 2021 (with <10 sequences) *GISAID, 2008*. However, the comparative analysis on genome sampling frequencies revealed that expansion of B.1.617.2+only occurred after April 2021 (*Figure 1b*). We further confirmed that by August 2021, the lineage had reached a relative frequency of >95%, coinciding with the peak of the third wave of infection recorded in the country *Subsecretaria de Prevención y Promoción de la Salud, 2021a*. Up to the sampling date of this study, we detected >80 B.1.617.2 sub-lineages (AY.X) circulating in Mexico, with most viral genomes assigned as AY.20 (22%), AY.26 (13%), and AY.100 (5%), followed by AY.113, AY.62, and AY.3. Of interest, these were previously noted to be mostly prevalent within North America (https://cov-lineages.org/) *O'Toole et al., 2021g*; *O'Toole et al., 2021h*; *O'Toole et al., 2021i*; *O'Toole et al., 2021j*; *O'Toole et al., 2021k*. During its circulation period, B.1.617.2+displayed a more homogeneous genome sampling distribution across Mexico, as compared to other virus lineages. Again, this is likely to be associated with the establishment of a systematic viral genome sampling and sequencing following February 2021, further driven by the widespread expansion of the lineage throughout the country (*Figure 4*).

Phylodynamic analysis for B.1.617.2+revealed a minimum number of 142 introduction events into Mexico (95% HPD interval = [125-147]). Within the MCC, six major clades denoting extended transmission chains were identified (C1d-C6d), with C1d, C3d, C5d, and C6d being the largest (*Figure 4*, *Supplementary file 2*). At least four independent introduction events were detected as the earliest (and synchronous) MRCAs, all dating to April 2021 (including the ancestral nodes of the C3d, C4d, and C6d clades). Based on inferred node dates in the MCC tree, we report a total lineage persistence of seven months (up to November 30, 2021). C2d comprises 'Mexico' virus genomes assigned as AY.62, sampled mainly from the state of Yucatán. Clade C4d comprises genomes from Mexico assigned as AY.3, sampled mostly from the central and south of the country (JAL) (*Figure 4*). Of interest, the C1d and C3d clades represent two independent introduction and spread events of the AY.26 sub-lineage into the country. C1d comprises genomes from Mexico sampled from the north (>60%; BCS, SIN, JAL), followed by central (CMX) and south-eastern states (VER, ROO, and YUC) (*Figure 4*). The MRCA of the C1d dates to May 2021, denoting a clade persistence of 6 months (from May 2021 to November 2021). Comparably, the C3d comprises genomes from Mexico mostly sampled from the north (37%; SIN, BCS, and SON). Comparably, the MRCA of the C3d dates to April 2021, denoting a clade persistence of 7 months (from April 2021 to November 2021) (*Figure 4*).

For the largest clades identified, C5d comprises viral genomes assigned as AY.100 (44%), to the parental B.1.617.2 (40%), and to the AY.113 (12%). Within this clade, we observe that the AY.100 and B.1.617.2 genomes are separated by a central sub-cluster of AY.113-assigned genomes (*Figure 4*). Approximately 70% of the genome sequences within C5d were sampled from Mexico (mostly assigned as AY.100 and AY.113), whilst 30% were sampled from the USA (mostly assigned as B.1.617.2). The majority of the 'Mexico' genomes are positioned basally and distally within the clade, sampled from all 32 states, but predominantly from north, centre and southern regions (>50%; represented by CHH, DUR, NLE, CMX, MEX, CAM, YUC, TAB, CHP, and ROO; *Figure 4*). Thus, the C5d represents the most genetically diverse and geographically widespread clade identified in Mexico. The MRCA of the C5d dates to May, denoting a clade persistence of up to 6 months (from May 2021 to November 2021). C6d is the second largest clade identified, comprising viral genomes from Mexico assigned as AY.20, mostly collected from central region of the country (>60%; represented by CMX, MEX, MOR, MIC, and HID) (*Figure 4*). Thus, contrasting to C5d, C6d denotes an extended transmission chain with a geographic distribution mainly restricted to central Mexico. The MRCA of the C6d clade dates to April, displaying a clade persistence of seven months (from April 2021 to November 2021).

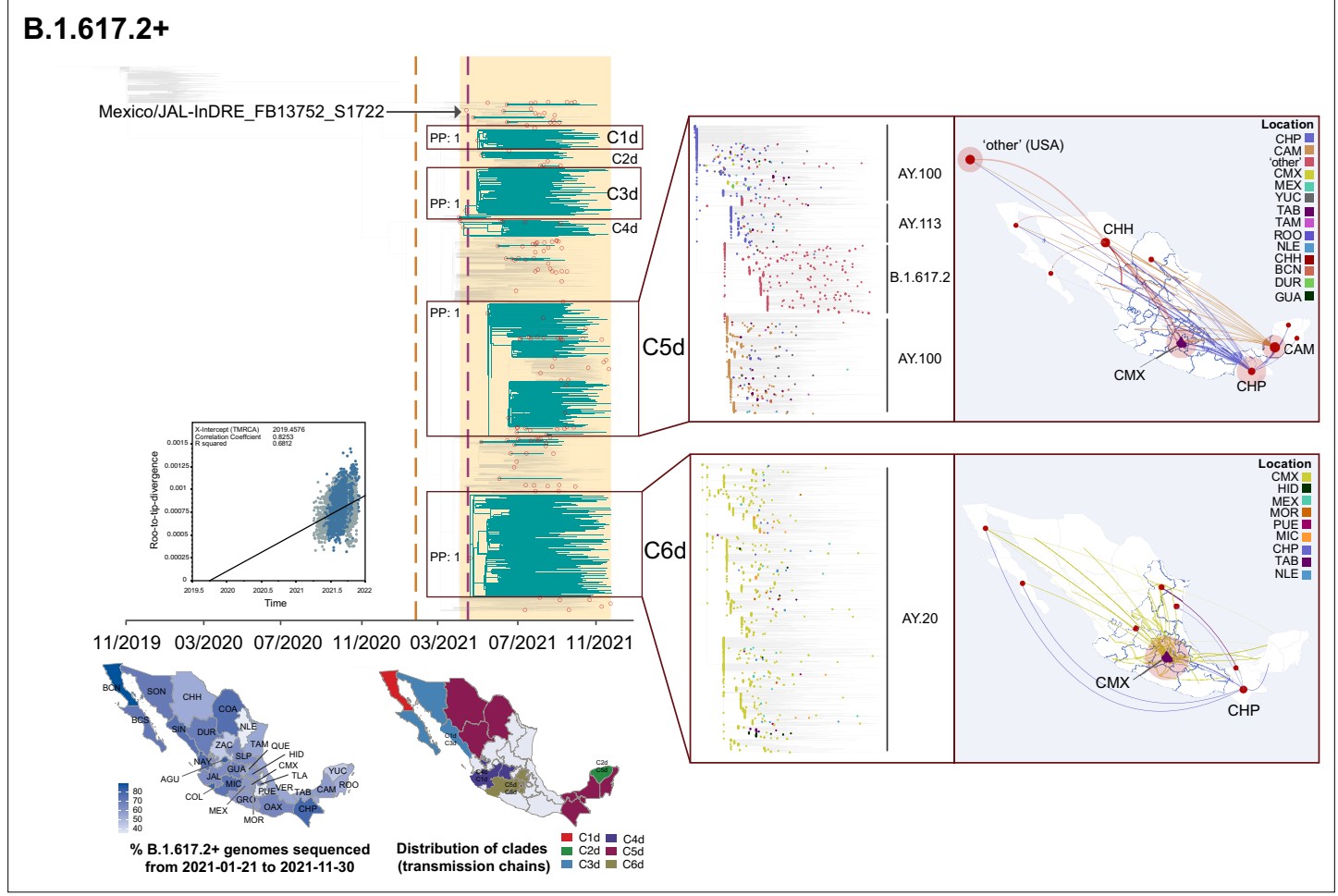

**Figure 4.** Time-scaled and phylogeographic analysis for the B.1.617.2 lineage. Maximum clade credibility (MCC) tree for the B.1.617.2 lineage, in which major clades identified as distinct introduction events into Mexico are highlighted. Nodes shown as red outline circles correspond to the most recent common ancestor (MRCA) for clades representing independent introduction events into Mexico. Based on the earliest and latest MRCAs, the estimated circulation period for each lineage is highlighted in yellow shadowing. The dashed purple line represents the date of the earliest viral genome sampled from Mexico, while its position in the tree is indicated. The dashed yellow line represents the implementation date of a systematic virus genome sampling and sequencing scheme for the surveillance of SARS-CoV-2 in Mexico. The corresponding root-to-tip regression plot for the tree is shown, in which genomes sampled from Mexico are shown in blue, whilst genomes sampled elsewhere are shown in grey. The map graph on the left show the cumulative proportion of genomes sampled across states per lineage of interest, corresponding to the period of circulation of the given lineage (relative to the total number of genomes taken from GISAID, corresponding to raw data before subsampling). The map on the right represents the geographic distribution of the main clades identified (for further details see *Supplementary file 2*). On the right, a zoom-in to the C5d and C6d clades showing sub-lineage composition with the most likely location estimated for each node. Geographic spread across Mexico inferred for these clades is further represented on the maps on the right, derived from a discrete phylogeographic analysis (DTA, see Methods section "Time-scaled analysis"). Viral transitions between Mexican states are represented by curved lines colored according to sampling location, showing only well-supported transitions (Bayes Factor >100 and a PP >0.9) (see *Table 1*).

The online version of this article includes the following video(s) for figure 4:

**Figure 4—video 1.** Animated visualizations of the spread pattern inferred for the C5d clade across Mexico derived from the DTA phylogeographic analysis.

https://elifesciences.org/articles/82069/figures#fig4video1

**Figure 4—video 2.** Animated visualizations of the spread pattern inferred for the C6d clade across Mexico derived from the DTA phylogeographic analysis.

https://elifesciences.org/articles/82069/figures#fig4video2

## Spread of B.1.617.2

Given the size and diversity of the C5d and C6 clades, we further explored viral diffusion patterns across the country using a phylogeographic approach (see Methods section "Time-scaled analysis"). For the C5d clade, viral spread is likely to have occurred from the south (represented by the states of Chiapas and Campeche; CHP and CAM) into the rest of the country (*Figure 4—video 1*). Well-supported transitions (scored under a BF >100 and a PP >.90) were mostly inferred from the southern state of Campeche (CAM) into central and northern states, and subsequently from the northern state of Chihuahua (CHH) into the central and northern region of the country, with some bidirectionality observed. Well-supported transitions were also observed from Baja California into Chihuahua (BCN/BCS into CHH), and from Chihuahua into the USA (arbitrarily represented by the geographic coordinates of the state of California) (*Figure 4*). Contrastingly, for C6d, a limited viral spread was observed from central states (represented by Mexico City, CMX) into central, northern and southern regions of the country (again with some bidirectionality observed). Well-supported transitions were also inferred from the southern state of Chiapas into central and northern region of the country (*Figure 4—video 2*). Bayes Factor (BF) and Posterior Probability (PP) for well-supported transitions observed between locations can be found as *Table 1*.

## Linking virus spread to human mobility data

Human movement can directly contribute to virus spread into unexposed areas, while mobility patterns may also reveal the impact of social and demographic factors on epidemics, such as population density and the effectiveness of non-pharmaceutical interventions at different scales. Thus, tracking human mobility is crucial to understand virus spread, especially when other factors cannot fully explain observed trends. Our analysis on human mobility data derived from mobile phone usage (collected between January 2020 and December 2021 at a national scale, see Methods section "Human mobility data analysis and exploring correlations with genomic data"), revealed two mobility peaks across time (*Figure 1c*). The first occurred between February and April 2021, coinciding with the introduction and spread of the B.1.1.7 and P.1+lineages, and with the contraction of B.1.1.519. The second mobility peak was observed between August and November 2021, coinciding with the expansion of the B.1.617.2 lineage. Increased human movement (represented by the cumulative number of trips into a given state) were observed for Mexico City, and to a lesser extent, for Jalisco, the State of Mexico, Nuevo León and Puebla (followed by Coahuila, Guanajuato, and Veracruz; *Figure 1c*). Mean connectivity within national territory revealed intensified movements from Mexico City into the State of Mexico, Morelos, Hidalgo, Puebla, Veracruz and Jalisco, and from Jalisco into Michoacán and Guadalajara (*Figure 1—figure supplement 2*). However, for both the C5d and C6d clades, no correlation between viral transitions and mean connectivity was observed (C5d: Adjusted R-squared: 0.006577, F-statistic: 7.15 on 1 and 928 DF, p-value: 0.007628. C6d: Adjusted R-squared: 0.3216, F-statistic: 470.8 on 1 and 990 DF, p-value:<2.2e-16), nor with the pairwise distance between states (C5d: Adjusted R-squared: 0.01086, F-statistic: 4.051 on 1 and 277 DF, p-value: 0.04511. C6d: Adjusted R-squared: 0.02296, F-statistic: 2.715 on 1 and 72 DF, p-value: 0.1038). Many of the lineage-specific clades we identify displayed a geographic distribution within southern region of the country (i.e. clades C3a and C7 for B.1.1.7, clade C1_P1_17 for P.1, and clades C2d and C5d for B.1.617.2). In this context, ranking connectivity between the southern region of the country (represented by Yucatán, Quintana Roo, Chiapas and Campeche) and the remaining 28 states did reveal a consistently high number of bidirectional movements between regions (represented by the CAM, CMX, and VER; *Figure 1—figure supplement 2*).

## Discussion

Our results reveal contrasting epidemiological and evolutionary dynamics between virus lineages circulating in Mexico during the first year of the epidemic, with some identifiable patterns. Both the B.1.1.222 and B.1.1.519 lineages likely originated in Mexico, characterized by single clades denoting extended sustained transmission for over a year. During this time, both lineages dominated in Mexico, and were seeded back and forth between Mexico and the USA, but never dominated across the USA. Thus, the number of publicly available viral genomes from each country reflects sequencing disparities that contrast with the lineage-specific epidemiological patterns observed across regions, highlighting

**Table 1.** Bayes Factor (BF) and Posterior Probability (PP) for well-supported transitions observed between locations*.

| C5d | | | | C6d | | | |
|-----|-----|-----|-----|-----|-----|-----|-----|
| Location | | | | Location | | | |
| From | To | BFR | PP | From | To | BFR | PP |
| BCN | CHH | 14535.32494 | 1 | AGU | CHP | 13635.15617 | 1 |
| CAM | CHP | 14535.32494 | 1 | BCN | CHP | 13635.15617 | 1 |
| CAM | CMX | 14535.32494 | 1 | CHP | CMX | 13635.15617 | 1 |
| CAM | MEX | 14535.32494 | 1 | CHP | COA | 13635.15617 | 1 |
| CAM | MIC | 14535.32494 | 1 | CHP | DUR | 13635.15617 | 1 |
| CAM | other | 14535.32494 | 1 | CHP | GRO | 13635.15617 | 1 |
| CAM | QUE | 14535.32494 | 1 | CHP | GUA | 13635.15617 | 1 |
| CAM | ROO | 14535.32494 | 1 | CHP | HID | 13635.15617 | 1 |
| CAM | SLP | 14535.32494 | 1 | CHP | JAL | 13635.15617 | 1 |
| CAM | SON | 14535.32494 | 1 | CHP | MEX | 13635.15617 | 1 |
| CAM | TAB | 14535.32494 | 1 | CHP | MIC | 13635.15617 | 1 |
| CAM | TAM | 14535.32494 | 1 | CHP | NLE | 13635.15617 | 1 |
| CAM | TLA | 14535.32494 | 1 | CHP | OAX | 13635.15617 | 1 |
| CAM | VER | 14535.32494 | 1 | CHP | other | 13635.15617 | 1 |
| CAM | ZAC | 14535.32494 | 1 | CHP | PUE | 13635.15617 | 1 |
| CMX | CHH | 14535.32494 | 1 | CHP | QUE | 13635.15617 | 1 |
| CHH | CHP | 14535.32494 | 1 | CHP | SIN | 13635.15617 | 1 |
| CHH | CMX | 14535.32494 | 1 | CHP | SLP | 13635.15617 | 1 |
| CHH | DUR | 14535.32494 | 1 | CHP | SON | 13635.15617 | 1 |
| CHH | GUA | 14535.32494 | 1 | CHP | TAB | 13635.15617 | 1 |
| CHH | MIC | 14535.32494 | 1 | CHP | TLA | 13635.15617 | 1 |
| CHH | NLE | 14535.32494 | 1 | CHP | VER | 13635.15617 | 1 |
| CHH | QUE | 14535.32494 | 1 | CAM | CHP | 13635.15617 | 0.998890122 |
| CHH | TAB | 14535.32494 | 1 | NLE | TAB | 13635.15617 | 0.998890122 |
| CHH | TAM | 14535.32494 | 1 | CHP | TAM | 6810.002999 | 0.997780244 |
| CHH | VER | 14535.32494 | 1 | CHP | YUC | 2714.911095 | 0.99445061 |
| CHH | ZAC | 14535.32494 | 1 | MEX | PUE | 164.4591205 | 0.915649279 |
| CAM | CMX | 14535.32494 | 0.998890122 | | | | |
| CHH | TLA | 14535.32494 | 0.998890122 | | | | |
| CAM | SIN | 3621.718465 | 0.995560488 | | | | |
| BCS | CHH | 1023.240732 | 0.984461709 | | | | |
| MIC | YUC | 468.8988157 | 0.966703663 | | | | |
| CHH | other | 399.6060762 | 0.961154273 | | | | |
| CAM | COA | 188.7999953 | 0.921198668 | | | | |
| MEX | YUC | 126.5111615 | 0.886792453 | | | | |

*derived from the phylogeographic analyses for C5d and C6d (B.1.617.2+). Only values of BF >100 and PP >0.9 are shown.

the need to leverage genomic surveillance efforts across neighboring nations using joint strategies (*Hill et al., 2021*).

Further similarities were observed for the P.1 and B.1.1.7 lineages, for which peaks in genome sampling frequencies coincided with a decrease in cases following the second wave of infection. We further confirm that P.1 and B.1.1.7 did not dominate in Mexico *Taboada et al., 2021*; *Zárate et al., 2022*, in contrast to what was observed in other countries such as the UK and USA (*Vöhringer et al., 2021*; *Faria et al., 2021*; *Washington et al., 2021*). Similar observations were independently made for other Latin American countries (some with better genome representation than others, like Brazil *Faria et al., 2021*), suggesting that the overall epidemiological dynamics of B.1.1.7 in Latin America may have differed substantially from that observed in the USA and UK (*Vöhringer et al., 2021*; *Faria et al., 2021*; *Washington et al., 2021*). Such differences could be explained partly by competition between lineages, exemplified in Mexico by the regional co-circulation of B.1.1.7, P.1, and B.1.1.519. Nonetheless, a lack of representative number of viral genomes for most of these countries prevents exploring such hypothesis at a larger scale, and further highlights the need to strengthen genomic epidemiology-based surveillance. However, the overall evolutionary dynamics observed for these VOC are comparable to those reported in other countries (*Vöhringer et al., 2021*; *Faria et al., 2021*; *Washington et al., 2021*). As an example, in the USA, the earliest introductions reported for the B.1.1.7 lineage were synchronous to those observed in Mexico (occurring between October and November 2020) and were also characterized by a few extended transmissions chains with a distribution often constrained to specific states (*Washington et al., 2021*). Thus, as drawn from our study, the successful spread of a given virus lineage does not seem to be linked to a higher number of introduction events, but rather to the extent and distribution of transmission chains, with size likely reflecting virus transmissibility (*Grenfell et al., 2004*).

In Mexico, the introduction and spread of the B.1.1.7, P.1, and B.1.617.2 lineages was characterized by multiple introduction events, resulting in a few successful extended local transmission chains. The epidemiological and evolutionary dynamics of the three VOC show that not only did these coincide temporally, but also revealed multiple and independent transmission chains corresponding to different lineages (and sub-lineages) spreading across the same geographic regions. Our results further revealed several clades belonging to different virus lineages distributed within the south region of the country, suggesting that this area has played a key role in the spread of SARS-CoV-2. Of notice, such pattern is comparable to what has been observed for arboviral epidemics in Mexico (*Gutierrez et al., 2023*; *Thézé et al., 2018*). Despite differences in the transmission mechanism between SARS-CoV-2 and arboviruses, we speculate that common epidemiological patterns may have emerged in Mexico due to the dependence of vector populations on human behavior and mobility patterns (*Gutierrez et al., 2023*; *Thézé et al., 2018*). Virus transmission rates may also vary within specific regions due to population density coupled with social factors (for example, unregulated migration across borders). Jointly, these observations indicate that the southern region of Mexico (represented by the states of Chiapas, Yucatán, and Quintana Roo) may be a common virus entry and seeding point, emphasising the need for an enhanced virus surveillance in states that share borders with neighboring countries, and highlights the importance of devising social behavior-informed tailored surveillance strategies applied to specific states (i.e. sub-region-specific surveillance).

In general, an increasing growth rate (*Rt*; defined as the instantaneous reproductive number that measures how an infection multiplies *Royal Statistical Society, 2021*) observed for different SARS-CoV-2 lineages dominating across specific regions can be partially explained by a fluctuating virus genetic background (i.e. emerging mutations that impact viral fitness) (*Vöhringer et al., 2021*; *Faria et al., 2021*; *Washington et al., 2021*). In light of our results, relative to the parental B.1.1.222 lineage, B.1.1.519 displayed only two amino acid changes within the Spike protein: T478K and P681H (*Rodríguez-Maldonado et al., 2021*). Mutation T478K locates within the Receptor Binding Domain (RBD), with a potential impact in antibody-mediated neutralization (*Liu et al., 2021*; *Wilhelm et al., 2021*). On the other hand, mutation P681H is located upstream the furin cleavage site, and falls within an epitope signal hotspot (*Haynes et al., 2021*). Thus, it may enhance virus entry (*Lubinski et al., 2021*), reduce antibody-mediated recognition (*Haynes et al., 2021*), and confer Type I interferon resistance (*Lista et al., 2021*). We speculate that at least one of these mutations may have contributed to the dominance of B.1.1.519 over B.1.1.222 by increasing the *Rt*, as has been observed for other SARS-CoV-2 subpopulations (*Martin et al., 2021*; *Escalera-Zamudio et al., 2023*; *Dolan et al., 2018*).

In agreement with this observation, in Mexico, an *Rt* of 2.9 was estimated for B.1.1.519 compared to an *Rt* of 1.93 estimated for B.1.1.222 (both calculated during epidemic week 2020–46, coinciding with the early expansion of B.1.1.519 in the country) (*Cedro-Tanda et al., 2021*).

Notably, the mutations observed for B.1.1.519 were not exclusive to the lineage, as P681H emerged later and independently in B.1.1.7 (and to a lesser extent in P.1, corresponding to 5% of all sampled genomes) (*Naveca et al., 2021*; *Hodcroft, 2021b*), whilst mutation T478K subsequently appeared in B.1.617.2 (*Liu et al., 2021*; *Lubinski et al., 2021*; *Lista et al., 2021*). Although assessing the impact of emerging mutations on lineage-specific fitness requires experimental validation, data derived from the natural virus population evidences amino acid changes at site 681 of the Spike protein have been predominantly fixed in VOC, with some mutations likely to yield an evolutionary advantage (*Liu et al., 2021*; *Lubinski et al., 2021*; *Lista et al., 2021*; *Hodcroft and Neher, 2021*; *Pond, 2021*; *Hodcroft, 2021a*). Thus, we propose that a somewhat 'shared' genetic background between the B.1.1.519 and B.1.1.7 lineages (as represented by mutation P681H) may have limited the spread of B.1.1.7 across the country. In this context, our finding suggest that the specific dominance and replacement patterns observed in Mexico were driven (to some extent) by lineage-specific mutations impacting growth rate, with competition between virus lineages at a local scale playing an important role.

Nonetheless, lineage-specific replacement and dominance patterns are likely to be shaped by the immune landscape of the local host population (*Gupta and Anderson, 1999*). In Mexico, relatively widespread and constant exposure levels to genetically similar virus subpopulations for extended periods of time (represented by the B.1.1.222 and B.1.1.519 lineages) may have yielded consistently increasing immunity levels in a somewhat still naïve population (with a nationwide seroprevalence of ~33.5% estimated by December 2020 *Basto-Abreu et al., 2022*; *Muñoz-Medina et al., 2021*). As more genetically divergent virus lineages were introduced and began to spread across the country (represented by B.1.1.7 and P.1, and later by B.1.617.2), a shift in the local immune landscape is likely to have occurred, impacted by a viral genetic background prompting (a partial) evasion of the existing immune responses. Supporting this observation, a vaccination rate of above 50% was only reached after December 2021 (*Ritchie, 2020*; *Bhatia et al., 2020*), suggesting that immunity levels during the first year of the epidemic mostly depended on virus pre-exposure levels.

In the context of human movement related to the spread of SARS-CoV-2 in Mexico, the fluctuating mobility patterns we observed for the country were consistent with a decrease in cases following the second and third waves of infection, likely reflecting changes in the color-coded system regulating travel restrictions leveraged by the risk of infection (*Gobierno de México, 2021b*). However, contrasting to our expectations on viral diffusion processes to be associated with local human mobility patterns, the geographic distances and overall human mobility trends observed within Mexico did not correlate with the virus diffusion patterns inferred (represented by B.1.617.2). As geographic distances and human mobility cannot be considered potential predictors of SARS-CoV-2 spread in Mexico, viral diffusion could be explained (to some extent) by human movement across borders. Taking this into consideration, it has been proposed that the spread of SARS-CoV-2 in Mexico is linked to human mobility across USA (for example, see *Zárate et al., 2022*), as we further evidence in this study by the transmission patterns observed for the B.1.1.222 and the B.1.1.519 lineages at an international scale. However, some of the virus diffusion patterns we observed are also congruent with human migration routes from South and Central America, supporting the notion that SARS-CoV-2 spread in Mexico has been impacted by epidemics within neighboring regions, and further underlines the need to investigate the potential role of irregular migration on virus spread across geographic regions (*Kraemer et al., 2020b*; *IOM Global Migration Data Analysis Centre, 2021*; *Varela Huerta, 2018*; *París-Pombo, 2016*).

Limitations of our study include uncertainty in determining source locations for virus introduction events into the country (for most lineages), restricted by regional genome sampling biases (*Kalkauskas et al., 2021*; *De Maio et al., 2015*). This is further impacted by (*i*) an uneven genome sampling across foreign locations and within the country, and (*ii*) by a poor viral genome representation for many countries in Central and South America (*GISAID, 2008*; *Gutierrez et al., 2021*). Such biases are also likely to affect the viral diffusion reconstructions we present, likely rendering them incomplete. However, as SARS-CoV-2 genome sampling and sequencing in Mexico has been sufficient, we are still able to robustly quantify and characterize lineage-specific transmission chains. It is worth highlighting that a differential proportion of cumulative viral genomes sequenced per state does not necessarily

mirror the geographic distribution and extension of the transmission chains identified, but rather represents a fluctuating intensity in virus genome sampling and sequencing through time. This underscores the importance of conducting phylogenetic inference-based analyses to explore viral spread, as opposed to relying solely on estimates derived from genome data frequency across time and space. A more homogeneous sampling across the country is unlikely to impact our main findings, but could (*i*) help pinpoint additional clades we are currently unable to detect, (*ii*) provide further details on the geographic distribution of clades across other regions of the country, and (*iii*) deliver a higher resolution for the viral spread reconstructions we present. Overall, our study prompts the need to better understand the impact of land-based migration across national borders, and encourages joint virus surveillance efforts in the Americas.

## Methods

### Data collation and initial sequence alignments

Global genome datasets assigned to each 'Pango' (*Rambaut et al., 2020*) lineage under investigation (B.1.1.222, B.1.1.519, B.1.1.7, P.1, and B.1.617.2) were downloaded with associated metadata from the GISAID platform (https://www.gisaid.org/) as of November 30th 2021 (*GISAID, 2008*; *Shu and McCauley, 2017*). The total number of sequences retrieved for each virus lineage were the following: B.1.1.222=3461, B.1.1.519=19,246, B.1.1.7=913,868, P.1=87,452, and B.1.617.2=2,166,874. These also included all SARS-CoV-2 genomes from Mexico available up the sampling date of this study, generated both by CoViGen-Mex and by other national institutions. Viral genome sequences were quality filtered to be excluded if presenting incomplete collection dates, if >1000 nt shorter than full genome length, and/or if showing >10% of sites coded as ambiguities (including N or X). Individual datasets were further processed using the Nextclade pipeline to filter according to sequence quality (*Aksamentov et al., 2021*). In addition, a set of the earliest SARS-CoV-2 sequences sampled from late 2019 to early 2020 (including reference sequence Wuhan-Hu-1, GenBank accession ID: MN908947.3), and a set of viral genomes representing an early virus diversity sampled up to May 31, 2020 were added for rooting purposes (https://github.com/BernardoGG/SARS-CoV-2_Genomic_lineages_Ecuador; *Gutierrez, 2021*). To generate whole genome alignments, datasets were mapped to the reference sequence Wuhan-Hu-1 (GenBank: MN908947.3) using Minimap2 (*Li, 2018*). Then, the main viral ORFs (Orf1ab and S) were extracted to generate reduced-length alignments of approximately 25,000 bases long, comprising only the largest and most phylogenetically informative coding genome regions (excluding smaller ORFs, UTRs, and short intergenic sequences).

### Migration data and phylogenetically-informed subsampling

To provide an overview for global introductory events into Mexico as a proxy for dataset reduction, we used openly available data describing anonymized relative human mobility flow into different geographical regions based on mobile data usage (*Kraemer et al., 2020a*; *Inward et al., 2022*) (https://migration-demography-tools.jrc.ec.europa.eu/data-hub/index.html?state=5d6005b30045242cabd750a2). For any given dataset, all 'non-Mexico' sequences were sorted according to their location, selecting only the top 5 countries representing the most intense human mobility flow into Mexico. In the case reported sub-lineages, the subsampled datasets were further reduced by selecting the top 5 sub-lineages that circulate(d) in the country. The 'Mexico' genome sets were then subsampled to 4000 in proportion to the total number of cases reported across time (corresponding to the epidemiological weeks from publicly accessible epidemiological data from the country *Gobierno de México, 2021a*). This yielded datasets of a maximum of 8,000 genomes, with an approximate 1:1 ratio of 'Mexico' to 'globally sampled' viral genomes (keeping those corresponding to the earliest and latest collection dates, sampled both from Mexico and globally). Preliminary Maximum Likelihood (ML) tress were then inferred using IQ-TREE (command line: iqtree -s -m GTR +I + G -alrt 1000 *Minh et al., 2020*).

Phylogenetically informed subsampling is based on maintaining basic clustering patterns, whilst reducing the noise derived from overrepresented sequences. This approach was applied to the ML trees resulting from the abovementioned migration-informed subsampled datasets, by using a modified version of Treemmer v0.3 (https://github.com/fmenardo/Treemmer/releases; *Menardo, 2019*; *Menardo et al., 2018*) to reduce the size and redundancy within the trees with a minimal loss of

diversity (*Menardo et al., 2018*). For this, the -lm command was initially used to protect 'Mexico' sequences and those added for rooting purposes. During the pruning iterations, the -pp command was used to protect 'Mexico' clusters and pairs of 'non-Mexico' sequences that are immediately ancestral or directly descending from these. This rendered reduced-size representative datasets that enable local computational analyses. As a note, clades may appear to be smaller relative to the raw counts of genomes publicly available, but actually reflect the sampled viral genetic diversity. Datasets were then used to re-estimate the ML trees, and used an input for time-scaled phylogenetic analysis (see Methods section "Time-scaled analysis"). Our subsampling pipeline is publicly accessible at (https:// github.com/rhysinward/Mexico_subsampling, copy archived at *Inward, 2022*).

We further sought to validate our migration-informed genome subsampling scheme (applied to B.1.617.2+, representing the best sampled lineage in Mexico). For this, an independent dataset was built using a different migration sub-sampling approach, comprising all countries represented by B.1.617.2+sequences deposited in GISAID (available up to November 30, 2021). In order to compare the number of introduction events, the new dataset was analysed independently under a time-scaled DTA (as described in Methods Section "Time-scaled analysis"). The distribution plots for each genome dataset before and after applying our migration- and phylogenetically informed subsampling pipeline, and a full description of the approach employed to validate our migration-informed subsampling is available as **Appendix 1**.

## Dataset assembly for initial phylogenetic inference

Given the reduced size of the original B.1.1.222 dataset, all sequences retained after initial quality filtering were used for further analyses. This resulted in a 3849-sequence alignment (including 760 genomes from Mexico). All other datasets (B.1.1.519, B.1.1.7, P.1, and B.1.617.2) were processed under the pipeline described (in Methods section "Migration data and phylogenetically-informed subsampling") to render informative datasets for phylogeographic analysis. The B.1.1.519 final dataset resulted in a 5001-sequence alignment, including 2501 genomes from Mexico. The B.1.1.7 final dataset resulted in a 7049-sequence alignment, including 1449 genomes from Mexico. The P.1 final dataset resulted in a 5493-sequence alignment, including 2570 genomes from Mexico. The B.1.617.2 final dataset resulted in a 5994-sequence alignment, including 3338 genomes from Mexico. All genome sequences used are publicly available and are listed in *Supplementary file 1*. Individual datasets were then used for phylogenetic inference as described above, with the resulting trees inputted for a time-scaled analyses.

## Time-scaled analysis

Output ML trees were assessed for temporal signal using TempEst v1.5.3 (*Rambaut et al., 2016*), removing outliers and re-estimating trees when necessary. The resulting trees were then time-calibrated informed by tip sampling dates using TreeTime (*Sagulenko et al., 2018*) (command line: treetime -aln --tree --clock-rate 8e-4 --dates --keep-polytomies --clock-filter 0). Due to a low temporal signal, a fixed clock rate corresponding to the reported viral evolutionary rate estimated ($8 \times 10^{-4}$ substitutions per site per year) was used (*Su et al., 2020*; *MacLean et al., 2020*). Root-to-tip regression plots for the ML trees (prior to time calibration, and excluding rooting sequences) show a weak temporal signal, and support the use of a fixed molecular clock rate ($8 \times 10^{-4}$) for the temporal calibration of phylogenetic trees (*Figures 2–4*).

In order to quantify lineage-specific introduction events into Mexico and to characterize clades denoting local extended transmission chains, the time-calibrated trees were utilized as input for a discrete trait analysis (DTA), also known as discrete phylogeographic inference. This analysis enables to infer well-supported MRCAs (*most recent common ancestor*, referring to the node where a given trait is most likely to have originated) and the corresponding descending clades. DTA analyses were performed using BEAST v1.10.4 to generate maximum clade credibility (MCC) trees (*Dellicour et al., 2021*; *Suchard et al., 2018*; *Lemey et al., 2009*). A DTA approach was suitable for all cases, as only a few discrete locations relatively well sampled across time were considered (*Lemey et al., 2009*). Using fixed 'time-calibrated' trees as an input for the DTA is an effective way of circumventing the restrictions of computationally expensive analyses on large datasets (*Dellicour et al., 2021*). Although this approach allows to infer dated introduction events into the study area, it does not consider phylogenetic uncertainty. Thus, the most recent common ancestor 'MRCA' dates we report come without

credibility intervals. For all introduction events identified, the mean and associated HPD interval were assessed. Following a similar strategy as described in *du Plessis et al., 2021* 'Mexico' clades were identified as those composed by a minimum of two sister 'Mexico' viral genome sequences directly descending from another 'Mexico' sequence. Extended local transmission chains were identified as clades composed by >20 viral genome sequences, with at least 80% of these sampled from Mexico, and with ancestral nodes supported by a PP value of >.80. Based on the MCC trees, we further estimated 'total persistence' times for the lineages studied, defined as the 'interval of time elapsed between the first and last inferred introduction events associated with the MRCA of any given clade from Mexico'. On the other hand, the lag between the earliest introduction event (MRCA) and the earliest sampling date for any given lineage corresponds to a 'cryptic transmission' period.

For the B.1.1.7, P.1, and B.1.617.2 datasets, analyses were performed to estimate the number of transitions into Mexico from other (unknown) geographic regions. Thus, two locations were considered: 'Mexico' and 'other'. For the B.1.1.222 and B.1.1.519 datasets, we estimated the number of transitions between Mexico and the USA, based on the fact both these lineages were considered endemic to North America (with >90% of the virus genomes sampled from the USA and Mexico; *O'Toole et al., 2021b*). For this cases, three distinct geographic locations were considered: 'Mexico', 'USA' and 'other'. The 'most likely' locations for lineage emergence were further obtained by comparing relative posterior probabilities (PP) between inferred ancestral locations for the given TMRCAs (*Dellicour et al., 2021*; *Suchard et al., 2018*; *Lemey et al., 2009*). For all analyses, independent Monte Carlo Markov Chain (MCMC) were run for $10^6$ iterations, sampling every $10^3$ states. To assess for sufficient effective sample size values (i.e. ESS >200) associated with the estimated parameters, we inspected MCMC convergence and mixing using Tracer 1.7 (*Rambaut et al., 2018*). In the case of B.1.617.2, we further explored viral diffusion patterns across the country by running two additional DTAs applied to the largest monophyletic clades identified within the MCC tree (C5d and C6d). For this, we used 33 distinct sampling locations (including all 32 states from Mexico, plus an 'other' location, referring to viral genomes sampled from outside the country). Visualization of the viral diffusion patterns was performed using SpreadViz (https://spreadviz.org/home), an updated web implementation of the Spatial Phylogenetic Reconstruction of Evolutionary Dynamics software SpreaD3 (*Bielejec et al., 2016*). In order to identify well-supported transitions between locations (*Lemey et al., 2009*), SpreadViz was also further used to estimate Bayes Factor (BF) values.

## Human mobility data analysis and exploring correlations with genomic data

Human mobility data used for this study derived from anonymized mobile device locations collected between 01/01/2020 and 31/12/2021 within national territory, made available by the company Veraset (*Fontanelli, 2020*). The source dataset includes anonymized identifiers for mobile devices, geographical coordinates (latitude and longitude) and a timestamp. The dataset was used to construct aggregated inter-state mobility networks, where nodes are defined as each of the 32 states from the country, whilst (weighed and directed) edges represent the normalized volume of observed trips between nodes (*Fontanelli, 2020*). The resulting networks were then used to quantify the number of cumulative trips from any state into a given specific state across time, the geographic distances among states, the mean inter-state connectivity observed between April 2021 and November 2021 (corresponding to the expansion period for the B.1.617.2 lineage, see *Figure 4b*, *Supplementary file 3*), and finally, for ranking connectivity between the south region of the country (represented by the states of Yucatán, Quintana Roo, Chiapas and Campeche) and the remaining 28 states (*Supplementary file 3*). The connectivity measure was defined as the sum of the weights for edges that go from any given node into other node(s), reflecting the number of trips in any direction. We then used the 'PhyCovA' software tool (https://evolcompvir-kuleuven.shinyapps.io/PhyCovA/) to perform preliminary analysis for exploring the human mobility data from the country as a potential predictor of viral transition among locations (*Blokker et al., 2022*). 'PhyCovA' was chosen as an explanatory approach over a fully-integrated GLM implemented in the Bayesian BEAST framework, as the last one would imply a high computational burden related to our datasets (*Suchard et al., 2018*).

## Acknowledgements

HGCS is supported by funding through the "Vigilancia Genómica del Virus SARS-CoV-2 en México" grant from the National Council for Science and Technology-México (CONACyT). SD acknowledges support from the Fonds National de la Recherche Scientifique (F.R.S.-FNRS, Belgium; grant n°F.4515.22), from the Research Foundation - Flanders (Fonds voor Wetenschappelijk Onderzoek-Vlaanderen, FWO, Belgium; grant n°G098321N), and from the European Union Horizon 2020 project MOOD (grant agreement n°874850). MEZ is currently supported by Leverhulme Trust ECR Fellowship (ECF-2019–542). OGP acknowledges support of the Oxford Martin School. MK and RPDI acknowledge support from the European Union Horizon 2020 project MOOD (#874850). The contents of this publication are the sole responsibility of the authors and do not necessarily reflect the views of the European Commission. The mobility team [MHR, AM, OF, MF, PG, GO, GAJ] and AHE are supported by 'Fondo Conjunto de Cooperación México-Uruguay' (Agencia Mexicana de Cooperación Internacional para el Desarrollo). CFA acknowledges support from grants "Vigilancia Genómica del Virus SARS-CoV-2 en México-2022" (PP-F003) from the National Council for Science and Technology-México (CONACyT), grant 057 from the "Ministry of Education, Science, Technology and Innovation (SECTEI) of Mexico City", and grant "Genomic surveillance for SARS-CoV-2 variants in Mexico" from the AHF Global Public Health Institute at the University of Miami. AL work was supported by DGAPA-PAPIIT (IN214421) and DGAPA-PAPIME (PE204921) of UNAM. We thank Verity Hill, Philippe Lemey, Tim Blokker and Sam Hong for their valuable advice on the technical details related to the methodology used for the time-scaled analyses. We thank all members of the Consorcio Mexicano de Vigilancia Genómica (CoViGen-Mex) for their efforts on sample collection and generating genetic sequence and metadata. Particularly, we thank León Martínez-Castilla and José Campillo Balderas for their contributions in the initial collation of preliminary data. We gratefully acknowledge all data contributors for the GISAID sequence data: i.e. the authors and their originating laboratories responsible for obtaining the specimens, and their submitting laboratories for generating the genetic sequence and metadata and sharing via the GISAID initiative, on which this research is based.

Zoom-in on the C3a and C7a clades identified as the largest within the B.1.1.7 MCC tree. Branch sampling locations are indicated only for sub-clusters composed of >5 sequences. The C3a clade is composed of 254 genome sequences sampled from 22/32 states in the country, mostly from Mexico City (CMX) and State of Mexico (MEX). Clade C7a is composed of 364 genome sequences, sampled mostly from the southern state of Tabasco (TAB). For details of all genome sequences within each clade see *Supplementary file 2*.

Zoom-in on the C1_P1 and C1_P1_17 clades, identified as the largest within the P.1+MCC tree. Branch sampling locations are indicated for large sub-clusters composed of >5 sequences. The C1_P1 clade is composed of 277 genome sequences, mostly sampled from the central region of Mexico City (CMX). The C1_P1_17 clade is composed of 588 genome sequences, mostly sampled from the southern states of Quintana Roo (ROO) and Yucatán (YUC). For details of all genome sequences within each clade see *Supplementary file 2*.

# Additional information

### Funding

| Funder | Grant reference number | Author |
|---|---|---|
| FNRS | F.4515.22 | Simon Dellicour |
| Research Foundation Flanders | G098321N | Simon Dellicour |
| European Horizon 2020 project MOOD | 874850 | Simon Dellicour |
| Leverhulme Trust | ECF-2019-542 | Marina Escalera Zamudio |
| CONACyT "Vigilancia Genómica del Virus SARS-CoV-2 en México-2022" | PP-F003 | Carlos F Arias |

| Funder | Grant reference number | Author |
|---|---|---|
| Ministry of Education, Science, Technology and Innovation (SECTEI) of Mexico City | 057 | Carlos F Arias |
| AHF Global Public Health Institute at the University of Miami | "Genomic surveillance for SARS-CoV-2 variants in Mexico" | Carlos F Arias |
| UNAM | DGAPA-PAPIIT (IN214421 | Antonio Lazcano |
| UNAM | DGAPA-PAPIME (PE204921) | Antonio Lazcano |

The funders had no role in study design, data collection and interpretation, or the decision to submit the work for publication.

### Author contributions

Hugo G Castelán-Sánchez, Luis Delaye, Formal analysis, Methodology, Writing – review and editing; Rhys PD Inward, Natalia Martinez de la Vina, Guillermo de Anda Jáuregui, Data curation, Methodology, Writing – review and editing; Simon Dellicour, Formal analysis, Validation, Investigation, Writing – review and editing; Bernardo Gutierrez, Formal analysis, Methodology; Celia Boukadida, Data curation, Visualization, Methodology, Writing – review and editing; Oliver G Pybus, Supervision, Validation; Plinio Guzmán, Selene Zárate, Blanca Taboada, Data curation, Investigation, Writing – review and editing; Marisol Flores-Garrido, Gabriela Olmedo-Alvarez, Investigation, Writing – review and editing; Óscar Fontanelli, Data curation, Formal analysis, Investigation; Maribel Hernández Rosales, Data curation, Formal analysis; Amilcar Meneses, Data curation, Investigation; Alfredo Heriberto Herrera-Estrella, Investigation, Visualization, Methodology, Writing – review and editing; Alejandro Sánchez-Flores, José Esteban Muñoz-Medina, Andreu Comas-García, Data curation, Writing – review and editing; Bruno Gómez-Gil, Conceptualization; Susana López, Carlos F Arias, Conceptualization, Writing – review and editing; Moritz UG Kraemer, Conceptualization, Methodology, Writing – review and editing; Antonio Lazcano, Conceptualization, Investigation, Writing – review and editing; Marina Escalera Zamudio, Conceptualization, Data curation, Formal analysis, Supervision, Validation, Investigation, Methodology, Writing - original draft, Project administration, Writing – review and editing

### Author ORCIDs

Luis Delaye (ID) http://orcid.org/0000-0003-4193-2720
Rhys PD Inward (ID) https://orcid.org/0000-0003-0016-661X
Simon Dellicour (ID) http://orcid.org/0000-0001-9558-1052
Celia Boukadida (ID) http://orcid.org/0000-0002-1744-0083
José Esteban Muñoz-Medina (ID) https://orcid.org/0000-0002-1289-4457
Selene Zárate (ID) https://orcid.org/0000-0003-1034-204X
Blanca Taboada (ID) https://orcid.org/0000-0003-1896-5962
Carlos F Arias (ID) https://orcid.org/0000-0003-3130-4501
Marina Escalera Zamudio (ID) http://orcid.org/0000-0002-4773-2773

### Decision letter and Author response

Decision letter https://doi.org/10.7554/eLife.82069.sa1
Author response https://doi.org/10.7554/eLife.82069.sa2

---

## Additional files

### Supplementary files

• Supplementary file 1. Virus genome IDs and GISAID accession numbers for the sequences used in each dataset.

• Supplementary file 2. Full list of names of all genome sequences within each major clade identified for each virus lineage.

• Supplementary file 3. Mobility matrixes summarizing: 1. Ranking connectivity between the southern region of the country, 2. Pairwise distances between states, 3. Mean intrastate connectivity.

• Transparent reporting form

## Data availability

Virus genome IDs and GISAID accession numbers for the sequences used in each dataset are provided in the Supplementary file 1 file. All genomic and epidemiological data supporting the findings of this study is publicly available from GISAID/GenBank, from the Ministry Of Health Mexico (*Gobierno de Mexico - Dirección General de Epidemiología, 2021*), and/or from the 'Our World in Data' coronavirus pandemic web portal (*Ritchie, 2020*). For the GISAID data used, the corresponding acknowledgement table is available on the 'GISAID Data Acknowledgement Locator' under the EPI_SET_20220405qd and EPI_SET_20220215at keys (*GISAID, 2008*). Our bioinformatic pipeline implementing a migration data and phylogenetically-informed sequence subsampling approach is publicly available at https://github.com/rhysinward/Mexico_subsampling (copy archived at *Inward, 2022*).

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

# Appendix 1

## Description of the approach employed to validate our migration-informed subsampling

### Methods

Our migration-informed approach aims to mitigate the effects of geographical over-representation impacting phylogeographic analyses, in which some regions may appear more frequently as seeding sources than they actually are. Applied to the B.1.617.2 (representing the best sampled lineage in Mexico), we sought to further validate our migration-informed genome subsampling approach by analysing an independent dataset built using a different migration sub-sampling scheme. This new scheme now comprised sub-sampling from all countries represented by B.1.617.2+sequences deposited in GISAID (available up to November 30th 2021). For this, complete virus genome sequences assigned to the B.1.617.2+lineage collected globally up to November 30th 2021 available from GISAID (https://www.gisaid.org/) were downloaded as of December 26th 2022. Genome sequences were quality filtered and retained according to the criteria stated in Methods Section "Data collation and initial sequence alignments" (see main text). Genome sequences were further sub-sampled in order to obtain an equal and homogeneous spatial and temporal representation of all geographic regions represented by all the sequences downloaded from GISAID (*i.e.,* countries), keeping the number of sampled sequences proportional to the number of cases officially reported from Mexico (corresponding to the epidemiological weeks matching the circulation period of the B.1.617.2 lineage within the country). We further added the set of earliest SARS-CoV-2 sequences sampled globally ('ROOT' outgroup), and the 3,320 subsampled genome sequences available from Mexico (used in the original B.1.617.2+dataset, as described in Methods Section "Migration data and phylogenetically-informed subsampling", main text). The final dataset resulted in an alignment of 25,107 columns and 6,912 sequences. Subsampling resulted in a homogeneous representation of ≈ 70 countries with an equal number of sequences (10–80 genome per country) per country, relative to their representation in GISAID (*Figure 1*). From these, approximately 3,570 genome sequences were sampled from any other country (*i.e.,* 'global' sample), preserving the 1:1 sequence ratio of 'Mexico' vs 'non-Mexico' sequences. The resulting new dataset was further processed and analysed to infer the number of introduction events into Mexico (corresponding to MRCA nodes associated to independent 'Mexico' clades), following the steps described in Methods Section "Time-scaled analysis" (main text).

### Results and discussion

In the new dataset,<100 genome sequences from the USA were retained for further analysis (*Figure 1*), compared to approximately 2,000 genome sequences from the USA included in the original B.1.617.2+alignment. Thus, we expected a lower number of inferred introduction events into Mexico, as an under-sampling of viral genome sequences from the USA is likely to result in 'Mexico' clades not fully segregating (particularly impacting C5d). Our original results revealed a minimum number of 142 introduction events into Mexico (95% HPD interval = [125-148]), with 6 major clades identified (denoting extended transmission chains). The DTA results derived from the new dataset (subsampling all countries) revealed a minimum number of 85 introduction events into Mexico (95% HPD interval = [82-87]), with again 6 major clades identified (*Figure 2*).

Thus, a significantly lower number of introduction events into Mexico were inferred, as was expected. On the other hand, the number of clades identified were consistent between both datasets, supporting for the robustness of our phylogenetic methodological approach. However, in the new dataset, we observe that C5d displayed a reduced diversity, represented by the AY.113 and AY.100 genomes from Mexico, but excluded the B.1.617.2 genome sampled from the USA (as seen in *Taboada et al., 2021*). This highlights the relevance of our genome sub-sampling using migration data as a proxy. In further agreement with our observations, publicly available data on global human mobility (https://migration-demography-tools.jrc.ec.europa.eu/data-hub/index.html?state=5d6005b30045242cabd750a2) shows that migration into Mexico is mostly represented by movements from the USA, followed by Indonesia, Guatemala, Belize and Colombia. However, the volume of movements from the USA into Mexico is much higher (up to 6 orders of magnitude above the volumes recorded into Mexico from any other country). This further supports for our migration informed subsampling approach of selecting only the top 5 countries with the highest migration rate into Mexico.

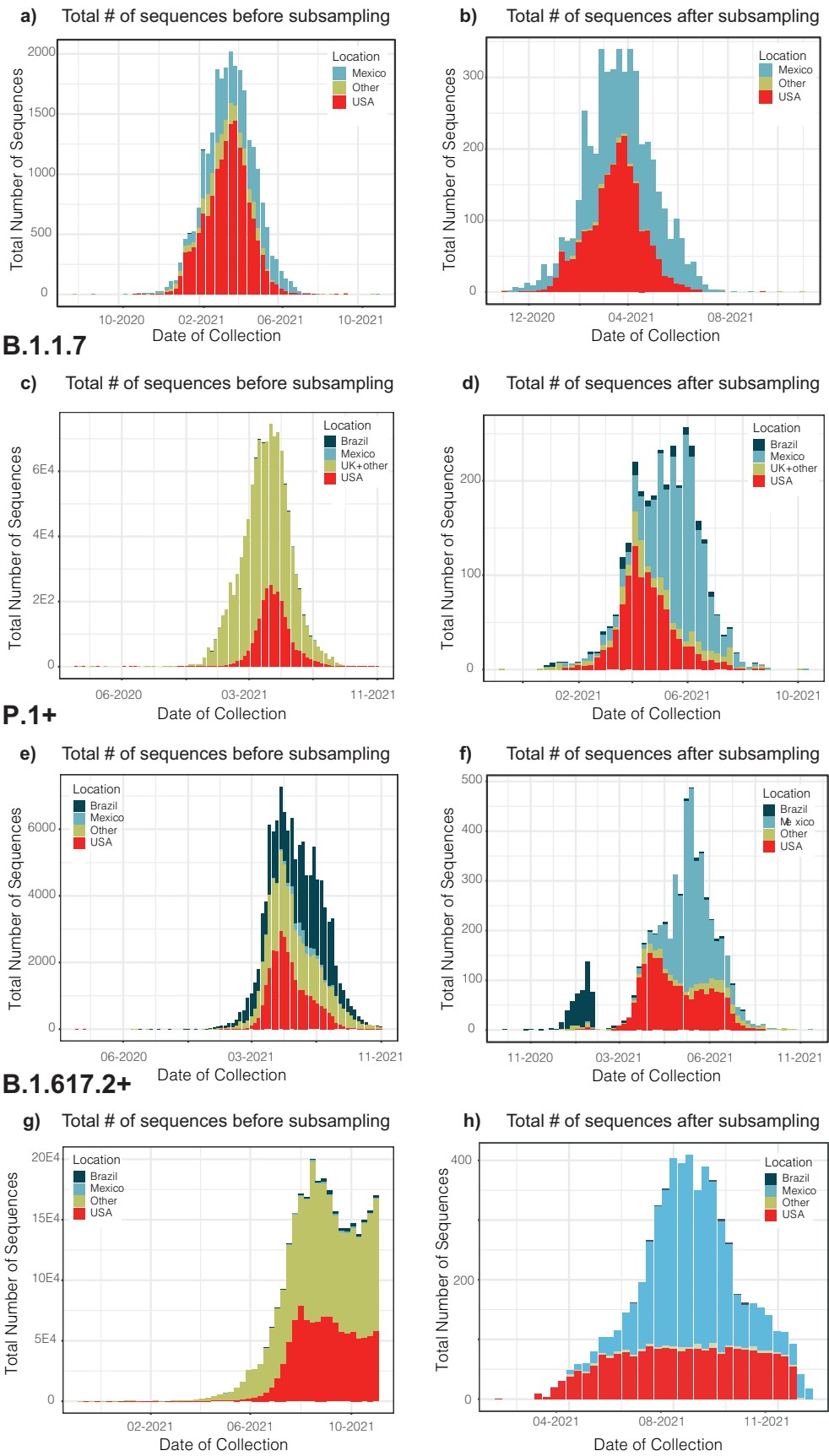

**Appendix 1—figure 1.** Distribution plots for each genome dataset before and after applying our migration- and phylogenetically-informed subsampling pipeline. Distribution plots for the number of genomes in the datasets before and after applying our subsampling pipeline. Plots for the B.1.1.519 (**a and b**), B.1.1.7 (**c and d**), P.1+ (**e and f**), and B.1.617.2+ (**g and h**) show the total number of sampled genomes colored according to location, ranked according to the countries representing the most intense human mobility flow into Mexico derived from anonymized relative human mobility flow into different geographical regions.

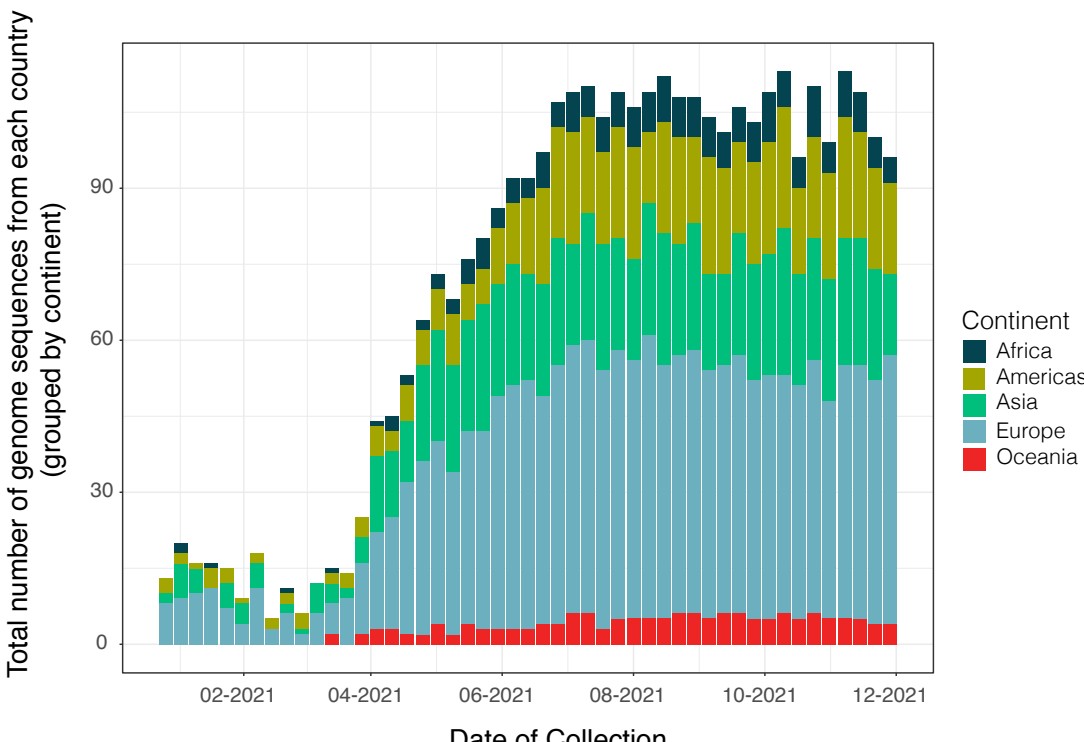

**Appendix 1—figure 2.** Distribution of genome sequences the new B.1.617.2+dataset after subsampling under a different migration-informed approach (validation). Distribution of the number of genomes in the dataset corresponding to an alternative sub-sample of B.1.617.2+sequences used for the validation of our migration informed subsampling approach. The dataset was built to obtain a homogeneous and proportional number of genome sequences from all countries sampled in GISAID (relative to their availability in the platform). The total number of genomes sequences sampled per region (represented by countries grouped by continent) are colored according to their continent of origin. To compare to the distribution of genome sequences before subsampling, see Appendix 1—figure 1 above.

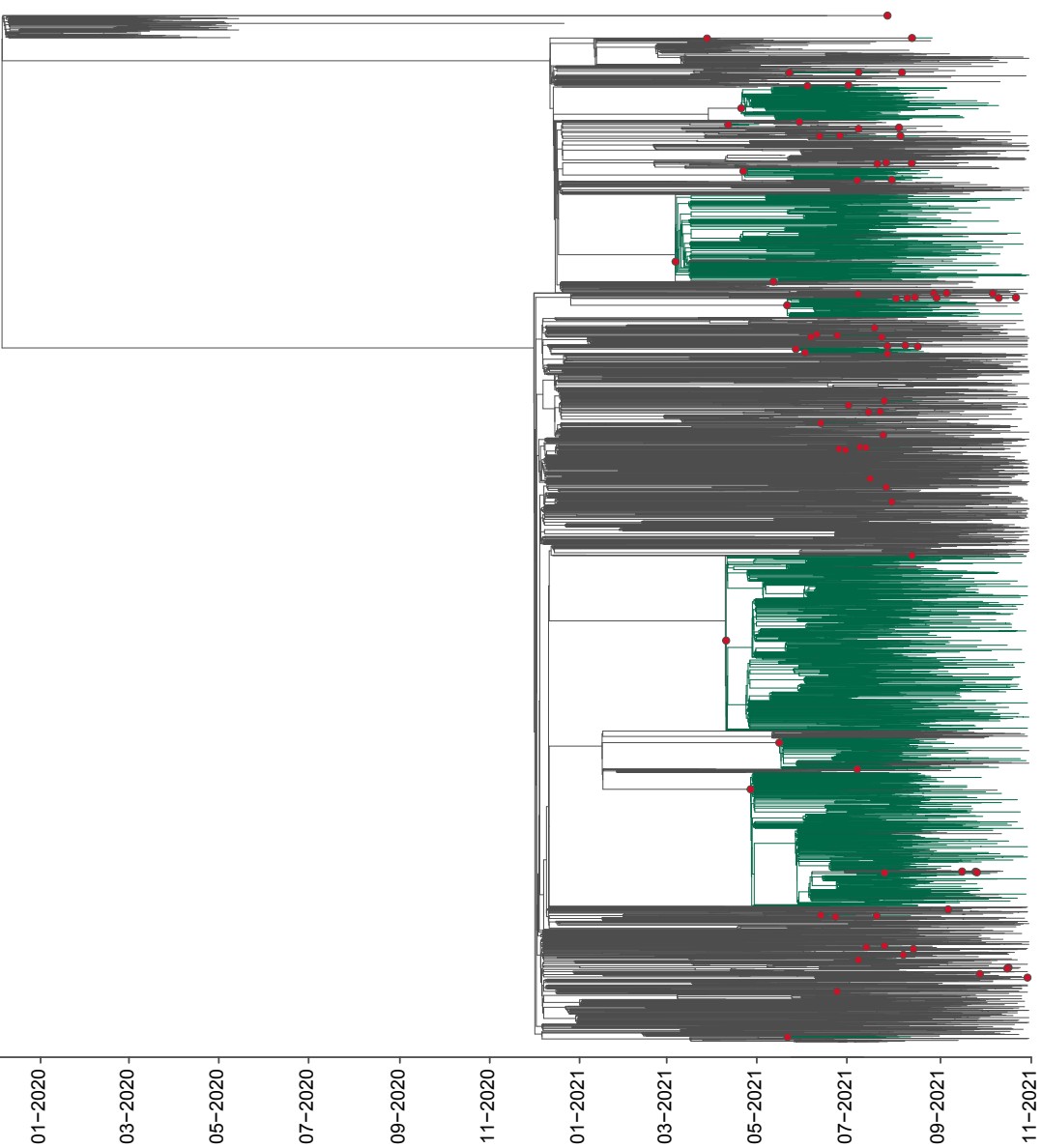

**Appendix 1—figure 3.** DTA analysis for the new B.1.617.2+dataset (validation). Maximum clade credibility (MCC) tree for the alternative B.1.617.2+dataset comprising a sub-sampling from all countries, represented by B.1.617.2+sequences deposited in GISAID available up to November 30th 2021, in which major clades identified as distinct introduction events into Mexico are highlighted. Nodes shown as red circles correspond to the inferred most recent common ancestor (MRCA) for clades representing independent introduction events into Mexico.

