## [Editor Report]

Castelán-Sánchez et al. analyzed, in an important way, the SARS-CoV-2 genomes from Mexico collected between February 2020 and November 2021. This period spans three major spikes in daily COVID-19 cases in Mexico and the rise of three distinct variants of concern (VOCs; B.1.1.7, P.1., and B.1.617.2). The authors perform convincing appropriate methodology in line with the current state of the art by careful phylogenetic analyses of these three VOCs, as well as two other lineages that rose to substantial frequency in Mexico, focusing on identifying periods of cryptic transmission (before the lineage was first detected) and introductions to and from the neighboring United States.

---

## [Decision Letter]

**Decision letter after peer review:**

Thank you for submitting your article "Comparing the evolutionary dynamics of predominant SARS-CoV-2 virus lineages co-circulating in Mexico" for consideration by *eLife*. Your article has been reviewed by 2 peer reviewers, and the evaluation has been overseen by a Reviewing Editor and Neil Ferguson as the Senior Editor. The reviewers have opted to remain anonymous.

Essential revisions:

1) The introduction should be modified so that the scientific rationale, questions being asked, and significance of the results are clear from the start. While many of these points become clear throughout the results and discussion, defining motivating hypotheses/questions in the introduction will improve readability.

2) Authors should provide details on the distribution of samples across the various Mexican States and over time. Although this information is provided in Supplementary Data 2, it is not presented in a way that enables the reader to evaluate if lineages were truly predominant in certain regions of the country, or if these results are attributable purely to sampling bias (see specific comments below)

3) It would be useful to include sensitivity analyses to show whether the decision to only include sequences from the top 5 countries of migration impact the main findings.

*Reviewer #1 (Recommendations for the authors):*

This is a generally well-written paper that presents phylogenetic analyses of specific SARS-CoV-2 lineages circulating in Mexico. However, as written, the manuscript is largely descriptive and is not motivated by any specific hypotheses or questions. There is also a limited connection between the key conclusions for different lineages and why they may be important for understanding or mitigating SARS-CoV-2 in Mexico. I think that re-writing the manuscript to focus on specific questions that can be answered with the analyses performed (some of which are currently briefly mentioned in the Discussion section) and the implication of these answers for outbreak control, SARS-CoV-2 mitigation, or some other purpose, would make the paper a stronger fit for a journal such as *eLife*.

Specific comments for consideration by the authors, many of which point out specific concerns about the effects of potential sampling bias on the conclusions:

A figure showing the number of sequences per region over time and additional information on what the specific findings mean in this context could be helpful in resolving some of the sampling bias concerns.

In general, the authors use "sampling frequency" to refer to the frequency of a specific lineage among all sequenced samples from a particular point in time. This is a highly confusing term, in my opinion, as the phrase is more typically used to refer to the proportion of cases that were sequenced at a particular time. Especially given some of my concerns about sampling bias, I would strongly suggest the authors be more careful about how they describe sampling, frequency, etc. throughout.

Another general comment is that it is sometimes unclear when the authors are presenting new data, and when they are citing a previous study, in part due to the large number of citations throughout. Could the authors work on rephrasing some of their findings so it is clear what is new, what is confirmatory, and what was published already?

Figure 1B. This is a small point, but why are daily cases plotted as a line graph and excess mortality as a series of small peaks? Is this due to the frequency of the collected data? Would it be appropriate to apply some sort of smoothing or sliding window approach?

Line 174: What exactly is impacted by an uneven genome sequencing across states? Being more specific about the effects of sampling bias could assuage some of the sampling concerns previously brought up.

Line 196: I was a bit confused to read about the "observed increase", as October 2020 does not correspond to a peak in cases in Mexico. There is indeed a small spike at that time, but it is unclear without additional data if this is truly due to increased transmission of the B.1.1.222 lineage, or if perhaps samples were collected more frequently during the relevant time and place.

Line 197: How do the authors know that frequency was initially underestimated for the B.1.1.222 lineage? Was sampling sufficient and sufficiently distributed across states that they can confidently state that their initial estimates were below the true frequency? The authors clearly recognize the importance of representative sampling – perhaps they can apply an existing method (e.g., https://www.medrxiv.org/content/10.1101/2021.12.30.21268453v1 or others) to provide some basis for their conclusions.

Line 204: I was a little confused about how the authors are inferring directionality between spread from the United States to Mexico or vice versa. Some of these analyses seem to be based on the fact that the lineage was observed in Mexico (as in this example for B.1.1.222) – when was this lineage first identified in the United States, and how sure are the authors that the lineage in fact emerged in Mexico before the United States?

Line 209: Is total persistence based on sample collection dates or phylogenetic analysis? From line 246 it seems that collection dates were used, though perhaps a phylogeny-based estimate would be more accurate, given the finding of cryptic transmission.

Line 219: The authors note that the B.1.1.222 outliers were from other countries. Why is this of note? Do the authors expect the variant to be different there? Sequencing errors? If the authors want to note this fact, I think they should explain why it is notable.

Line 315: Could this lag in detection also affect the frequencies of each lineage shown in Figure 1b?

Line 394: Where is the human mobility data from? Also, I imagine that the journal may require the figures to be referenced in order, in which case it may make sense to move the mobility data in Figure 1c to a figure later in the manuscript.

Line 399: Is the cumulative number of trips into a given state the most informative value for understanding movement, or would it be helpful to also/alternatively normalize this value by state population? Perhaps a cumulative number of trips makes sense if this value does not include trips within a state. Did the authors exclude trips that start and end within the same state?

*Reviewer #2 (Recommendations for the authors):*

1. The authors took great care in crafting their subsampling approach, and overall I think that the method makes a lot of sense. Balancing out/subsampling to even out the distribution of sequences from different countries makes sense, and is a good method for reducing bias. However, it would be useful to include a supplemental figure or 2 that evaluates how their decision to only include sequences from the top 5 countries of migration impact their results. I found myself wondering whether using only the top 5 countries impacts the total number of introductions they infer, and it might be nice to show that their general findings are robust to this choice. From my reading, the vast majority of analyses focus on using DTA to compare 2-3 groups: Mexico vs. not Mexico and Mexico vs. the USA vs. not Mexico. Because the number of groups is low, adding additional sequences in the "not Mexico" category from more countries should not make the computation intractable, as long as the overall number of genomes is kept similar. I would suggest the authors repeat one of the analyses (one of the variant trees) using 2 alternative subsampling: 1. including the top 10 countries of migration, and 2. including many many countries. In both, continuing to standardize the number of genomes per country would make sense. For the analysis with many many countries, this would likely require only keeping a few genomes per country, and I would guess would result in fewer inferred introductions into Mexico. However, this finding might be a nice way to show that you get a better resolution of introductions with the approach they chose. This is just a suggestion, and some other analysis may suffice, but I think that some additional figures and text showing the difference their chosen sampling regime makes to the final resulting tree would be great.

2. The paper is very carefully done and makes very reasonable conclusions about the data. However, I found myself wondering during my first read what questions the authors were hoping to answer and why. In the discussion, the authors articular some interesting points, and it became clear. I would suggest that the authors add more information into the introduction that lays out explicitly the scientific rationale and significance of their results. I think that this would help readers understand why this paper is interesting from the beginning better.

[Editors' note: further revisions were suggested prior to acceptance, as described below.]

Thank you for resubmitting your work entitled "Comparing the evolutionary dynamics of predominant SARS-CoV-2 virus lineages co-circulating in Mexico" for further consideration by *eLife*. Your revised article has been evaluated by Neil Ferguson (Senior Editor) and a Reviewing Editor.

The manuscript has been improved but there are some remaining issues that need to be addressed, to improve the clarity of the paper, as outlined below:

*Reviewer #1 (Recommendations for the authors):*

The authors have done a nice job of addressing my concerns about sampling across various Mexican states, and they appropriately mention sampling limitations throughout the manuscript. I appreciate the time they put into the revision and their commitment to careful analyses throughout. That said, I still have a few questions related to sampling that I would appreciate if the authors could address:

1. On Line 46, the authors state that "the program aimed to sequence per month 1,200 representative samples…" Did the program actually succeed in generating close to 1,200 SARS-CoV-2 sequences per month? There is limited information on exactly how many sequences were generated per region per time point (the newly added figure supplement addresses only proportions and does not divide sampling by time).

2. I think the second paragraph in the Results section (starting on Line 92) should come before the current first paragraph of the Results. I found it hard to interpret the results presented in this first paragraph until I read more details on the specific samples included.

3. Line 127: I think it will be important to address that 35% also likely isn't the true prevalence of the B.1.222 lineage. I think the authors agree with me based on their rebuttal ("As pointed out by the reviewer, it is true that both values may actually represent an underestimation of a 'real' lineage prevalence. However, as we have now clarified, the results we present are based on the fact that SARS-CoV-2 sequencing in Mexico has been sufficient to 'confidently' investigate the spatial and temporal distribution of viral lineages at a national scale. Moreover, our observations derive from currently available genome data, and thus calculating 'real' values is beyond our scope.") but this is not clear in the text. Furthermore, I think the point that the authors are trying to make regarding sequencing disparities across countries needs to be further spelled out (similarly, the comment in Line 167 about sequencing disparities also needs to be spelled out more clearly).

4. I'm a little bit confused by the maps included as part of Figure 2-4. In all cases, the left map seems to suggest that the lineage of interest was observed at some frequency in many different states, but the right map highlights only a small fraction of states. The figure legend says only that the maps on the right represent the geographic distribution of clades identified, but what are they actually showing? Is this a result from the phylogeographic analysis? How can a lineage be observed in many more regions (according to the map on the left) than included as part of the distribution on the map on the right?

5. Line 206: I think the authors should more clearly explain the B.1.1.7 findings in Chihuahua and mention the various explanations for their observations, including missing many cases, the fact that Chihuahua could be a frequent sink, etc. I believe the authors are considering all of these possibilities, but they are not well articulated in the revised manuscript.

The authors also improved the introduction to include a specific hypothesis, though I think this aspect could still be improved in a few small ways:

1. Line 58: Though I appreciate the authors adding in a hypothesis, the hypothesis presented is somewhat of an enumeration of various possible explanations for the fluctuating dominance of various SARS-CoV-2 lineages. Perhaps it would be clearer to state that there is a need to understand why some lineages rise to dominance and cause large numbers of cases, rather than framing them as a specific (or rather, not very specific) hypothesis. I do not think the authors need to use the word "hypothesis" in order to adequately set up the question(s) they are trying to answer.

2. Line 331: I think here is another place the authors could do a better job of explaining why they are performing specific analyses throughout the paper (not just in the introduction). For example, a short phrase at the top of the mobility data section beginning on Line 331, explaining how mobility could explain trends when other factors do not, would make the paper feel less purely descriptive.

Finally, I have a few small comments/questions that arose during the review of the revised manuscript:

1. Line 51: Over 80,000 genomes in GISAID as of what date?

2. Line 52: I am not sure if it is fair to claim that national institutions generated all of the sequences from Mexico. I know of at least some that were generated by labs in the United States (I ask this in the interest of accuracy, not because I don't think most genomes from Mexico should be generated in-country!).

3. Line 65: It would be helpful if the authors could indicate which of the listed Pango lineages were designated as VOCs (e.g., B.1.617.2 (Δ)).

4. Line 76: I believe "C5d" refers to a specific figure that hasn't been introduced yet. This reference was a little confusing without further context.

5. Line 132: This is a very minor point, but the authors frequently point out that various sequences derive from a single MRCA. However, this is true by definition, as any group of sequenced can be traced to an MRCA. I think the point the authors are trying to make is conveyed by the first part of the sentence, that B.1.222 sequences were part of one main clade (this comment applies to several other places in the manuscript as well).

6. Line 141: I previously asked if persistence was calculated using sample dates or phylogenetic analyses. The authors have clarified this, but it was only when I got to Line 246 that I actually understood what the authors were doing and why this was interesting/different than simply observing sequence dates. In general, the authors could do a better job throughout of making it clear why they are performing specific analyses (related to the first major comment above).

7. Line 154: The authors also addressed my question about the outliers in the tree. I appreciate the additional details, but the authors provide an oddly specific explanation for sequencing errors. Why do the sequencing errors need to come from using an inadequate reference sequence? It seems there could be many other possibilities, including contamination, low-quality sequences, etc. Could the authors just say that the outliers are likely to be due to sequencing errors and leave it at that?

8. Line 365: Peaks in genomic sampling frequencies correlating with a decrease in cases seems to suggest that the total sampling during those peak periods might have been lower (i.e., higher frequency because of a smaller denominator). Are there any results the authors could provide that show this was not the case (i.e., more details on temporal sampling)? Or if not, perhaps this is worth mentioning as a possibility explicitly?

9. Line 389: Why would the authors expect SARS-CoV-2 patterns to mirror arbovirus patterns, given very different mechanisms of transmission?

*Reviewer #2 (Recommendations for the authors):*

The authors have responded quite well to reviewer comments. My primary suggestion to the authors was to provide some additional analyses validating and exploring their migration-informed subsampling regime. The new data and text in the Appendix are excellent and fully address my questions/concerns. The authors also added a bit of text in the discussion and introduction adding some context to the purpose of these studies, which has improved the manuscript. I recommend publication. I did, however, find one typo:

Figure 2 legend: "Nodes are shown as red outline circles" – I think this is a typo, as there are no red circles in these trees that I could see.

---

## [Author Response]

Essential revisions:1) The introduction should be modified so that the scientific rationale, questions being asked, and significance of the results are clear from the start. While many of these points become clear throughout the results and discussion, defining motivating hypotheses/questions in the introduction will improve readability.2) Authors should provide details on the distribution of samples across the various Mexican States and over time. Although this information is provided in Supplementary Data 2, it is not presented in a way that enables the reader to evaluate if lineages were truly predominant in certain regions of the country, or if these results are attributable purely to sampling bias (see specific comments below)3) It would be useful to include sensitivity analyses to show whether the decision to only include sequences from the top 5 countries of migration impact the main findings.

We thank the reviewers and the editor for their constructive comments and acknowledge their value. We have incorporated all key changes suggested. We believe that this has improved the readability and quality of our manuscript, fulfilling the high-quality standards set by the *eLife* Journal. Changes include:

The main text has now been modified to frame the study within the hypothesis and goals stated. We have further aimed to connect key conclusions to public health implications, applied to the country and the extended geographic region of the Americas.

We now include a new figure (Supplementary Figure 1 c) comparing the overall cumulative proportion of genomes generated per state between 2020-2021. Furthermore, we have also added additional maps representing the geographic distribution of the clades identified within Figures 2-4. The main text has been modified to provide further details on the sampling scheme used by CoViGen-Mex, and we thoroughly discuss how this may impact our findings.

We now provide a comparative analysis to validate our migration-informed genome subsampling approach. For this, we re-ran the time-scaled analysis applied to the B.1.617.2 dataset (representing the best sampled lineage in the country) using a different migration-informed sub-sampling schemes (including all countries). We then re-estimated the number of introduction events and compared the obtained value with our initial result. This is now presented in a new section in SI (Supplementary Text 2 and Figure 29). For the new datasets, a significantly lower number of introduction events into Mexico were inferred whilst C5d displayed a reduced diversity, further supporting for our migration-informed subsampling approach.

We hold that our study and presents an important overview of different SARS-CoV-2 dynamics between developing and developed countries (represented by Mexico and the USA) and should be of interest to both local health and global authorities. We hope you consider our study for publication.

Reviewer #1 (Recommendations for the authors):This is a generally well-written paper that presents phylogenetic analyses of specific SARS-CoV-2 lineages circulating in Mexico. However, as written, the manuscript is largely descriptive and is not motivated by any specific hypotheses or questions. There is also a limited connection between the key conclusions for different lineages and why they may be important for understanding or mitigating SARS-CoV-2 in Mexico. I think that re-writing the manuscript to focus on specific questions that can be answered with the analyses performed (some of which are currently briefly mentioned in the Discussion section) and the implication of these answers for outbreak control, SARS-CoV-2 mitigation, or some other purpose, would make the paper a stronger fit for a journal such as eLife.Specific comments for consideration by the authors, many of which point out specific concerns about the effects of potential sampling bias on the conclusions:

The main text has now been modified to clearly state the hypotheses of the study (L59-65), and we now frame our findings in line with these. We have also aimed to connect key conclusions to public health implications, applied to both SARS-CoV-2 surveillance in Mexico and in the Americas. We hope the reviewer considers that these issues have now been addressed.

A figure showing the number of sequences per region over time and additional information on what the specific findings mean in this context could be helpful in resolving some of the sampling bias concerns.

To provide further details on ‘the distribution of samples across various Mexican states’, a new figure comparing the overall cumulative proportion of genomes generated per state between 2020-2021 is now available as Supplementary Figure 1 c (for further details, see the response to the Essential revisions).

In general, the authors use "sampling frequency" to refer to the frequency of a specific lineage among all sequenced samples from a particular point in time. This is a highly confusing term, in my opinion, as the phrase is more typically used to refer to the proportion of cases that were sequenced at a particular time. Especially given some of my concerns about sampling bias, I would strongly suggest the authors be more careful about how they describe sampling, frequency, etc. throughout.

We have defined the term ‘genome sampling frequency’ as follows: “the proportion of viral genomes assigned to a specific lineage, relative to the proportion of viral genomes assigned to any other virus lineage in a given particular time point” (L90-92). This now enables us to use the term in a standardized way throughout the main text. We hope the reviewer considers that the issue has been addressed.

Another general comment is that it is sometimes unclear when the authors are presenting new data, and when they are citing a previous study, in part due to the large number of citations throughout. Could the authors work on rephrasing some of their findings so it is clear what is new, what is confirmatory, and what was published already?

This has now been amended accordingly.

Figure 1B. This is a small point, but why are daily cases plotted as a line graph and excess mortality as a series of small peaks? Is this due to the frequency of the collected data? Would it be appropriate to apply some sort of smoothing or sliding window approach?

Excess mortality is represented by a punctuated grey curve denoting weekly average values. This has now been clarified in the corresponding figure legend. Highlighting the impact of COVID-19 for a given country is difficult because the reported numbers of cases and deaths can be strongly affected by testing capacity and case reporting. Thus, excess mortality is considered as an objective indicator of the for COVID-19 impact (see Karlinsky and Kobak 2021, *eLife* doi: 10.7554/*eLife*.69336). The purpose of showing the average number of daily cases together excess mortality was to provide a better overview of the impact of COVID-19 in Mexico during the first year of the epidemic. Given the illustrative purpose of the data, we do not consider necessary ‘to apply some sort of smoothing or sliding window approach’.

Line 174: What exactly is impacted by an uneven genome sequencing across states? Being more specific about the effects of sampling bias could assuage some of the sampling concerns previously brought up.Line 196: I was a bit confused to read about the "observed increase", as October 2020 does not correspond to a peak in cases in Mexico. There is indeed a small spike at that time, but it is unclear without additional data if this is truly due to increased transmission of the B.1.1.222 lineage, or if perhaps samples were collected more frequently during the relevant time and place.

This and related statement(s) have now been modified throughout the main text, in line with our ‘General response from authors’.

Line 197: How do the authors know that frequency was initially underestimated for the B.1.1.222 lineage? Was sampling sufficient and sufficiently distributed across states that they can confidently state that their initial estimates were below the true frequency? The authors clearly recognize the importance of representative sampling – perhaps they can apply an existing method (e.g., https://www.medrxiv.org/content/10.1101/2021.12.30.21268453v1 or others) to provide some basis for their conclusions.Line 204: I was a little confused about how the authors are inferring directionality between spread from the United States to Mexico or vice versa. Some of these analyses seem to be based on the fact that the lineage was observed in Mexico (as in this example for B.1.1.222) – when was this lineage first identified in the United States, and how sure are the authors that the lineage in fact emerged in Mexico before the United States?

To clarify the three points raised above, we have re-written the ‘B.1.1.222’ and ‘B.1.1.519’ Results sections.

Regarding the first point, the section has been amended as follows: “For the B.1.1.222, a rising genome sampling frequency was observed from May 2020 onwards, coinciding with the first epidemiological wave in the country recorded during July 2020. Subsequently, genome sampling frequency progressively increased to reach a highest of 35% recorded in October 2020, denoting an established dominance within the country before the emergence and spread of the B.1.1.519 (Figure 1b)” (L120-124).

As pointed out by the reviewer, it is true that both values may actually represent an underestimation of a ‘real’ lineage prevalence. However, as we have now clarified, the results we present are based on the fact that SARS-CoV-2 sequencing in Mexico has been sufficient to ‘confidently’ investigate the spatial and temporal distribution of viral lineages at a national scale. Moreover, our observations derive from currently available genome data, and thus calculating ‘real’ values is beyond our scope. Related to the second point raised, the ‘initial underestimation’ we mentioned is regarding the results analysed by others and published prior to our study. This has been clarified as follows: “Data corresponding to the first year of the epidemic available up to February 2021 (as analysed by Taboada et al. 2021) initially estimated that the B.1.1.222 lineage had reached a maximum genome sampling frequency of approximately 10%. Compared to our results, this revealed an important frequency underestimation (10% vs 35%), attributable to the fact that the vast majority of B.1.1.222-assigned genomes from Mexico (>80%) were generated, assigned and submitted to GISAID after February 2021” (L124-129).

For the third point, we consider important to mention that neither differences in the observed proportion of viral genomes sampled across locations (for example, more from the USA compared to Mexico), nor where or when a lineage was first detected can be used as a proxy to estimate the ‘most likely’ emergence location for any given lineage. Therefore, the sentence ‘analyses seem to be based on the fact that the lineage was observed in Mexico’ is incorrect. The earliest sampling locations and dates for the B.1.1.222 and the B.1.1.519 lineages (both in Mexico and globally) are stated in the corresponding Results section, but are somewhat irrelevant to the DTA results we present. In this sense, it is only by using a phylogeographic approach that we can examine the temporal and spatial congruency between inferred phylogenetic and geographic patterns using a statistical framework (see Lemey et al. 2009). This enables us to estimate a ‘most likely’ emergence locations for specific lineages/clades, by comparing the relative posterior probabilities between inferred ancestral locations associated with nodes representing most recent common ancestors (MRCAs). Thus, using this methodology, we can infer with high credibility that both the B.1.1.222 and B.1.1.519 lineages are emerged in Mexico, and that inference is unlikely to be affected by geographic sampling biases (at least for these two cases). As a note, inferring viral diffusion processes between specific locations could only be done for the B.1.1.222 and B.1.1.519, as these were considered endemic to North America (with >90% of all viral genomes coming from both the USA and Mexico). The corresponding results and methods sections have now been re-written to clarify this.

Line 209: Is total persistence based on sample collection dates or phylogenetic analysis? From line 246 it seems that collection dates were used, though perhaps a phylogeny-based estimate would be more accurate, given the finding of cryptic transmission.

This has now been clarified as follows: “Based on the MCC trees, we further estimated ‘total persistence’ times for the lineages studied, defined as the ‘interval of time elapsed between the first and last inferred introduction events associated with the MRCA of any given clade from Mexico’. On the other hand, the lag between the earliest introduction event (MRCA) and the earliest sampling date for any given lineage corresponds to a ‘cryptic transmission’ period.” (L560-564).

Line 219: The authors note that the B.1.1.222 outliers were from other countries. Why is this of note? Do the authors expect the variant to be different there? Sequencing errors? If the authors want to note this fact, I think they should explain why it is notable.

This has now been clarified as follows: “During our initial phylogenetic assessment, we noted that most of the B.1.1.519-assigned genomes collected after November 2021 came from outside North America (namely, from Turkey and Africa). These were further identified as outliers in the tree, likely to be sequencing errors resulting from the use of an inadequate reference sequence for genome assembly, and thus were excluded from further analyses.” (L154-158).

Line 315: Could this lag in detection also affect the frequencies of each lineage shown in Figure 1b?

The time-scaled inference approach we use provides a powerful tool for detecting incongruences between genome sampling dates vs. phylogenetic-derived observations. In this sense, only a poor virus genome representation (with no correlation observed between the number of sequences and cases reported), and/or gaps in sequencing efforts across time, and/or considerable lags between genome sequencing and uploading to GISAID (to make these publicly available) could affect lineage frequency estimations.

We do not expect any of these factors to have greatly impacted our results, and when a potential impact was noticed, it has been stated in the main text (see L154-159 and L246-249).

Line 394: Where is the human mobility data from? Also, I imagine that the journal may require the figures to be referenced in order, in which case it may make sense to move the mobility data in Figure 1c to a figure later in the manuscript.

The source of the human mobility data we use is originally stated in Methods section "Human mobility data analysis and exploring correlations with genomic data": “Human mobility data used for this study derived from anonymized mobile device locations collected between 01/01/2020 and 31/12/2021 within national territory, made available by the company Veraset.” (L586). The panel shown in Figure 1c results from our analysis of the data. Although it is true the figure is referenced later in the results, we hold it makes sense to keep it in its original format, as it provides an overall panorama of the results we present on SARS-CoV-2 spread in Mexico (including genomic and epidemiological data) in the context of human mobility patterns.

Line 399: Is the cumulative number of trips into a given state the most informative value for understanding movement, or would it be helpful to also/alternatively normalize this value by state population? Perhaps a cumulative number of trips makes sense if this value does not include trips within a state. Did the authors exclude trips that start and end within the same state?

The methodology we used is based on aggregated inter-state mobility networks, being one of the standard values for mobility data used to inform disease transmission modelling across geographic regions (see https://health.ec.europa.eu/system/files/2020-07/modelling_mobilitydata_en_0.pdf). Furthermore, this approach was previously used to explore the impact of human mobility linked to COVID-19 in Mexico (https://arxiv.org/abs/2203.13916v1).

We would like to clarify that we only used ‘inter-state’ mobility (and not ‘intra-state’). We want to rectify that ‘intra-state’ was mentioned by mistake, and this has now been corrected (L589 and L592). In this case, nodes represent municipalities and edges travels from municipality i to municipality j. The edge weight is the number of travels observed in a day from i to j, normalized by the number of devices observed that day. Such normalization allows us to compare networks between different days. Aggregated networks by state are built in order to have mobility networks where nodes represent states, edges travel between states, and weights the normalized number of travels between different states. We hope this we have made this clear now.

Reviewer #2 (Recommendations for the authors):1. The authors took great care in crafting their subsampling approach, and overall I think that the method makes a lot of sense. Balancing out/subsampling to even out the distribution of sequences from different countries makes sense, and is a good method for reducing bias. However, it would be useful to include a supplemental figure or 2 that evaluates how their decision to only include sequences from the top 5 countries of migration impact their results. I found myself wondering whether using only the top 5 countries impacts the total number of introductions they infer, and it might be nice to show that their general findings are robust to this choice. From my reading, the vast majority of analyses focus on using DTA to compare 2-3 groups: Mexico vs. not Mexico and Mexico vs. the USA vs. not Mexico. Because the number of groups is low, adding additional sequences in the "not Mexico" category from more countries should not make the computation intractable, as long as the overall number of genomes is kept similar. I would suggest the authors repeat one of the analyses (one of the variant trees) using 2 alternative subsampling: 1. including the top 10 countries of migration, and 2. including many many countries. In both, continuing to standardize the number of genomes per country would make sense. For the analysis with many many countries, this would likely require only keeping a few genomes per country, and I would guess would result in fewer inferred introductions into Mexico. However, this finding might be a nice way to show that you get a better resolution of introductions with the approach they chose. This is just a suggestion, and some other analysis may suffice, but I think that some additional figures and text showing the difference their chosen sampling regime makes to the final resulting tree would be great.

We have now included a validation analysis following the suggestions made. For further details, please see our ‘General response from authors’.

2. The paper is very carefully done and makes very reasonable conclusions about the data. However, I found myself wondering during my first read what questions the authors were hoping to answer and why. In the discussion, the authors articular some interesting points, and it became clear. I would suggest that the authors add more information into the introduction that lays out explicitly the scientific rationale and significance of their results. I think that this would help readers understand why this paper is interesting from the beginning better.

The main text has now been modified to clearly state the hypotheses of the study (L59-65), and to frame our finding in line with these. We have also aimed to connect key conclusions to public health implications, as applied to SARS-CoV-2 surveillance in Mexico and in the Americas. We hope the reviewer considers that these issues have now been addressed.

[Editors' note: further revisions were suggested prior to acceptance, as described below.]

The manuscript has been improved but there are some remaining issues that need to be addressed, to improve the clarity of the paper, as outlined below:

We thank the editor and the reviewer for their comments and positive feedback. Following this second revision process, we have incorporated all changes suggested to improve the clarity and readability of our study. Below you will find a point by point response addressing the issues raised by both reviewers, with line numbers corresponding to the tracked version of the manuscript.

Reviewer #1 (Recommendations for the authors):The authors have done a nice job of addressing my concerns about sampling across various Mexican states, and they appropriately mention sampling limitations throughout the manuscript. I appreciate the time they put into the revision and their commitment to careful analyses throughout. That said, I still have a few questions related to sampling that I would appreciate if the authors could address:1. On Line 46, the authors state that "the program aimed to sequence per month 1,200 representative samples…" Did the program actually succeed in generating close to 1,200 SARS-CoV-2 sequences per month? There is limited information on exactly how many sequences were generated per region per time point (the newly added figure supplement addresses only proportions and does not divide sampling by time).

We thank the Reviewer for raising this remark and have now completed the text as follows: “Derived from publicly accessible genome data from Mexico deposited in GISAID (https://www.gisaid.org/) from 2020 to 2021 (corresponding to the first year of the epidemic), over 48,000 viral genomes were available, resulting in an approximate average of 2,000 viral genomes sequenced per month. However, starting February 2021, the CoViGen-Mex sampling scheme gradually increased its sequencing output from a mean of less than 500 genomes per month to over 1000 genomes from May 2021 onwards [see plot below]. As of the time of writing this manuscript, around 80,000 SARS-CoV-2 genomes from Mexico were available in GISAID, with one-third generated by CoViGen-Mex ^30^ (and other national institutions sequencing the rest). From these, approximately 95% of all SARS-CoV-2 genomes from Mexico were generated in the country and 5% in the USA.” We have now incorporated this information in L51-59 accordingly. In general, an increasing sequencing pattern observed in Mexico, mirrors the mounting global sequencing efforts as the pandemic developed. We have now included the plot below as part of Figure 1 —figure supplement 1C. Additionally, all genome data statistics can be easily accessed through GISAID. We believe that this matter has now been adequately addressed.

2. I think the second paragraph in the Results section (starting on Line 92) should come before the current first paragraph of the Results. I found it hard to interpret the results presented in this first paragraph until I read more details on the specific samples included.

We have amended this accordingly (L146-157).

3. Line 127: I think it will be important to address that 35% also likely isn't the true prevalence of the B.1.222 lineage. I think the authors agree with me based on their rebuttal ("As pointed out by the reviewer, it is true that both values may actually represent an underestimation of a 'real' lineage prevalence. However, as we have now clarified, the results we present are based on the fact that SARS-CoV-2 sequencing in Mexico has been sufficient to 'confidently' investigate the spatial and temporal distribution of viral lineages at a national scale. Moreover, our observations derive from currently available genome data, and thus calculating 'real' values is beyond our scope.") but this is not clear in the text. Furthermore, I think the point that the authors are trying to make regarding sequencing disparities across countries needs to be further spelled out (similarly, the comment in Line 167 about sequencing disparities also needs to be spelled out more clearly).

As suggested, we have now addressed and clarified both issues as follows: “Data from the first year of the epidemic (available until February 2021, as analysed by Taboada et al., 2021^19,33,34^) initially revealed that the B.1.1.222 lineage had reached a maximum genome sampling frequency of approximately 10%. However, our results show an important frequency underestimation (10% vs 35%), since the vast majority of B.1.1.222-assigned genomes from Mexico (>80%) were generated, assigned, and submitted to GISAID after February 2021. These observations are based on publicly available genome data, and both these values may actually underestimate lineage prevalence. However, calculating 'real' frequency values goes beyond the scope of this study. Notably, in the USA, the B.1.1.222 lineage reached a maximum genome sampling frequency of 3.5%, compared to a 35% observed in Mexico. This four-fold difference in the number of B.1.1.222-assigned genomes between the USA and Mexico reflects a significant disparity in sequencing efforts between the two countries, and contrasts with region-specific epidemiological scenarios ^1^. This is of great importance since sequencing disparities and sampling gaps between countries can hinder the development of global outbreak control strategies and exacerbate existing health inequalities” (L207-219).

4. I'm a little bit confused by the maps included as part of Figure 2-4. In all cases, the left map seems to suggest that the lineage of interest was observed at some frequency in many different states, but the right map highlights only a small fraction of states. The figure legend says only that the maps on the right represent the geographic distribution of clades identified, but what are they actually showing? Is this a result from the phylogeographic analysis? How can a lineage be observed in many more regions (according to the map on the left) than included as part of the distribution on the map on the right?

Concerning our results, it is possible that the geographic distribution of a given viral lineage (derived from its sampling frequency [defined in L163], represented by the maps on the left), differs to the geographic distribution of clades belonging to the same virus lineage (denoting extended transmission chains within a specific region, represented by the maps on the right). This is because geographic distribution from sampling frequency is represented not only by sequences that form clades/clusters, but also by those that do not fall within any of these (namely singleton sequences/branches within a phylogenetic tree). On the other hand, the geographic distribution of clades can only be represented by sequences that exclusively fall within these (see Methods section "Time-scaled analysis"). Following the definition used for clade outlined in L748, these were preliminary identified within the time-calibrated trees, to be further confirmed as those displaying a well-supported MRCAs inferred under a discrete trait analysis (DTA; see response to minor point 5 below).

In this light, we had already discussed that “It is worth highlighting that a differential proportion of cumulative viral genomes sequenced per state does not necessarily mirror the geographic distribution and extension of the transmission chains identified, but rather represents a fluctuating intensity in virus genome sampling and sequencing through time” (L628-631). This underscores the importance of conducting phylogenetic inference-based analyses to explore viral spread, as opposed to relying solely on estimates derived from genome data frequency across time and space (now mentioned in L632-634). We hope this has now been clarified.

5. Line 206: I think the authors should more clearly explain the B.1.1.7 findings in Chihuahua and mention the various explanations for their observations, including missing many cases, the fact that Chihuahua could be a frequent sink, etc. I believe the authors are considering all of these possibilities, but they are not well articulated in the revised manuscript.

Following the previous review process, we explained that “during its circulation period, most B.1.1.7 genomes from Mexico were generated from the state of Chihuahua, with these also representing the earliest B.1.1.7-assigned genomes from the country ^34,44^. However, our analysis revealed that only a small proportion of these genomes grouped within a larger clade denoting an extended transmission chain (C2a), with the rest falling within minor clusters, or representing singleton events (Figure 3a). Relative to other states, Chihuahua generated an overall lower proportion of viral genomes throughout 2020-2021 (Figure 1 —figure supplement 1).” (L311-316). As requested, we have now added an additional paragraph discussing the possible role of Chihuahua in the spread/establishment of B.1.1.7 in Mexico:

“Between February 2021 and September 2021 (corresponding to the circulation period of the B.1.1.7 lineage/Α VOC in Mexico), Mexico City reported the highest number of COVID-19 cases (https://coronavirus.gob.mx/datos/#DOView). During this time, Mexico City also reported the highest number of cases related to the B.1.1.7 lineage/Α VOC, with 2,452 confirmed cases, followed by the states of Mexico, Jalisco, and Nuevo Leon (https://coronavirus.gob.mx/variantes/). Therefore, neither phylogenetic nor epidemiological data from the country support the notion that Chihuahua may have been an initial sink-source for the B.1.1.7 lineage/Α VOC (or for any other virus lineage, when comparing DTA results). Various factors can impact virus lineage distribution in a given region at a specific time point, including stochastic population processes, and the role of asymptomatic carriers, which can contribute to the difficulty in identifying extended transmission chains and their geographic distribution. Consequently, we can only speculate that given its proximity to the US border, Chihuahua may have been an early introduction point of the lineage from the US. However, this observation is not supported by our phylogeographic analyses, given the restrictions on determining source locations for virus introduction events into the country related to sampling limitations." (L317-330).

The authors also improved the introduction to include a specific hypothesis, though I think this aspect could still be improved in a few small ways:1. Line 58: Though I appreciate the authors adding in a hypothesis, the hypothesis presented is somewhat of an enumeration of various possible explanations for the fluctuating dominance of various SARS-CoV-2 lineages. Perhaps it would be clearer to state that there is a need to understand why some lineages rise to dominance and cause large numbers of cases, rather than framing them as a specific (or rather, not very specific) hypothesis. I do not think the authors need to use the word "hypothesis" in order to adequately set up the question(s) they are trying to answer.

The paragraph corresponding to the hypothesis of the study (L61-94) has been amended as follows: “Investigating the dominance and replacement patterns of SARS-CoV-2 can provide valuable information for understanding viral spread and shed light on virus evolution and adaptation processes. During the first year of the epidemic in Mexico, over 200 different virus lineages were detected, including all VOCs^19,31^. Various virus lineages co-circulated across the national territory, a noteworthy observation in light of recombinant SARS-CoV-2 lineages that emerged in North America during 2021 ^32^. Additionally, some virus lineages displayed specific dominance and replacement patterns that differed from those observed in neighbouring countries, specifically the USA ^30,33,34^. With this in mind, our study aimed to examine the dominance and replacement patterns of SARS-CoV-2 in Mexico from 2020 to 2021. We explored whether the spread of dominant lineages was driven by specific mutations that impacted local growth rates (further shaped the immune landscape of the local host population, depending mostly on virus pre-exposure levels at this time). We also investigated whether viral diffusion processes within the country were associated with local human mobility patterns, anticipating that the SARS-CoV-2 epidemic in Mexico may have been impacted by the epidemiological behaviour of the virus in neighbouring countries.”

2. Line 331: I think here is another place the authors could do a better job of explaining why they are performing specific analyses throughout the paper (not just in the introduction). For example, a short phrase at the top of the mobility data section beginning on Line 331, explaining how mobility could explain trends when other factors do not, would make the paper feel less purely descriptive.

We have now added a new statement at the beginning of the paragraph corresponding to this section (L460-464): “Human movement can directly contribute to virus spread into unexposed areas, while mobility patterns may also reveal the impact of social and demographic factors on epidemics, such as population density and the effectiveness of nonpharmaceutical interventions at different scales. Thus, tracking human mobility is crucial to understand virus spread, especially when other factors cannot fully explain observed trends.”

Finally, I have a few small comments/questions that arose during the review of the revised manuscript:1. Line 51: Over 80,000 genomes in GISAID as of what date?2. Line 52: I am not sure if it is fair to claim that national institutions generated all of the sequences from Mexico. I know of at least some that were generated by labs in the United States (I ask this in the interest of accuracy, not because I don't think most genomes from Mexico should be generated in-country!).

To address these two points, we have clarified the paragraph corresponding to L51-60 [see response to main point 1].

3. Line 65: It would be helpful if the authors could indicate which of the listed Pango lineages were designated as VOCs (e.g., B.1.617.2 (Δ)).

In L21-23 of the Introduction, we had provided a list of Pango lineages that have been designated as VOC. However, we have further amended L75-77 to explicitly state this as follows “To achieve this, we investigated the introduction, spread, and replacement dynamics of five virus lineages that dominated during the first year of the epidemic: B.1.1.222, B.1.1.519, B.1.1.7 (VOC Α), P.1 (VOC Γ), and B.1.617.2 (VOC Δ) ^30,33,34^.”

4. Line 76: I believe "C5d" refers to a specific figure that hasn't been introduced yet. This reference was a little confusing without further context.

We have clarified this sentence as follows: “For B.1.617.2, which represented the largest and most genetically diverse clades identified…” (L87-88).

5. Line 132: This is a very minor point, but the authors frequently point out that various sequences derive from a single MRCA. However, this is true by definition, as any group of sequenced can be traced to an MRCA. I think the point the authors are trying to make is conveyed by the first part of the sentence, that B.1.222 sequences were part of one main clade (this comment applies to several other places in the manuscript as well).

Following the point raised by the Reviewer, it is true that any group of related sequences within a tree can be traced to a given ancestral node. However, only under discrete trait analyses is that the MRCA refers to a well-supported node where a given trait (for example, a specific geographic location) is most likely to have originated. Therefore, an MRCA also represents the most recent point in time at which all sequences within a particular clade shared the same trait (see Duchêne et al., 2015. doi: 10.1371/journal.pcbi.1004550). Thus, in the context of our analytical approach, we hold that the use of the term ‘MRCA’ throughout the text is correct.

Nonetheless, we have aimed to clarify this in the sentence mentioned (L220-222), as follows: “Phylodynamic analysis for the B.1.1.222 lineage revealed one main clade deriving from a single earliest MRCA (most recent common ancestor) with a ‘most likely’ location (supported by a relative Posterior Probability [PP] of 0.99) inferred to be ‘Mexico’, denoting lineage emergence in the country”. Methods section "Time-scaled analysis" (L736-740) now also includes a brief definition of MRCA under DTA analyses: “In order to quantify lineage-specific introduction events into Mexico and to characterize clades denoting local extended transmission chains, the time-calibrated trees were utilized as input for a discrete trait analysis (DTA), also known as discrete phylogeographic inference. This analysis enables to infer well-supported MRCAs (most recent common ancestor, referring to the node where a given trait is most likely to have originated) and the corresponding descending clades.”

6. Line 141: I previously asked if persistence was calculated using sample dates or phylogenetic analyses. The authors have clarified this, but it was only when I got to Line 246 that I actually understood what the authors were doing and why this was interesting/different than simply observing sequence dates. In general, the authors could do a better job throughout of making it clear why they are performing specific analyses (related to point 1 above).

We had assumed that the Reviewer would notice the importance of our study's methodological approach, which emphasizes the significance of conducting phylogenetic inference-based analyses to explore virus spatial and temporal distribution. We have established the basis of this study on genomic epidemiology and believe that our methodological approach is sound and thoroughly explained in the corresponding section. We consider it unnecessary to explain the foundation and benefits of using phylodynamic inference to explore virus disease dynamics, as it goes beyond the scope of this study.

7. Line 154: The authors also addressed my question about the outliers in the tree. I appreciate the additional details, but the authors provide an oddly specific explanation for sequencing errors. Why do the sequencing errors need to come from using an inadequate reference sequence? It seems there could be many other possibilities, including contamination, low-quality sequences, etc. Could the authors just say that the outliers are likely to be due to sequencing errors and leave it at that?

We have amended this accordingly (L263-264).

8. Line 365: Peaks in genomic sampling frequencies correlating with a decrease in cases seems to suggest that the total sampling during those peak periods might have been lower (i.e., higher frequency because of a smaller denominator). Are there any results the authors could provide that show this was not the case (i.e., more details on temporal sampling)? Or if not, perhaps this is worth mentioning as a possibility explicitly?

When comparing the dates displayed on Figure 1b and Figure 1 —figure supplement 1C (also refer to the plot shown for Major point 1), it is evident that the total sampling during the peak periods between February 2021 and September 2021 (corresponding to the circulation period of the P.1 and B.1.1.7 lineages/Γ and Α VOCs in Mexico) was not decreasing but rather increasing. Additional information on temporal sampling has already been included in our revised version, and we have now made it clearer (please refer to Major point 1). We feel that this issue has been appropriately addressed.

9. Line 389: Why would the authors expect SARS-CoV-2 patterns to mirror arbovirus patterns, given very different mechanisms of transmission?

Arbovirus spread (in Mexico, and other regions of the world) is influenced by environmental conditions shaping the distribution of vector populations, but is also impacted by human behaviour and mobility (see Gutierrez et al. 2023, https://doi.org/10.1101/2023.03.08.22283959, reference 58 and Kraemer et al. 2019, https://doi.org/10.1038/s41564019-0376-y). Thus, “Despite differences in the transmission mechanism between SARS-CoV-2 and arboviruses, we speculate that common epidemiological patterns may have emerged in Mexico due to the dependence of vector populations on human behaviour and mobility patterns ^58,59^. Virus transmission rates may also vary within specific regions due to population density coupled with social factors (for example, unregulated migration across borders). Jointly, these observations indicate that the southern region of Mexico (represented by the states of Chiapas, Yucatán and Quintana Roo) may be a common virus entry and seeding point, emphasising the need for an enhanced virus surveillance in states that share borders with neighbouring countries, and highlights the importance of devising social behaviour-informed tailored surveillance strategies applied to specific states (i.e., sub region-specific surveillance).” (now discussed in now L521-532). We consider this remark of great importance, as it highlights the need to explore epidemiological convergence in independent virus populations, possibly driven by common social drivers.

Reviewer #2 (Recommendations for the authors):The authors have responded quite well to reviewer comments. My primary suggestion to the authors was to provide some additional analyses validating and exploring their migration-informed subsampling regime. The new data and text in the Appendix are excellent and fully address my questions/concerns. The authors also added a bit of text in the discussion and introduction adding some context to the purpose of these studies, which has improved the manuscript. I recommend publication. I did, however, find one typo:Figure 2 legend: "Nodes are shown as red outline circles" – I think this is a typo, as there are no red circles in these trees that I could see.

We thank the reviewer for noticing. We have amended this accordingly.